# Extremely Greedy Equivalence Search

**Achille Nazaret**[1]                          **David Blei**[1,2]

[1]Department of Computer Science, Columbia University, New York, USA
[2]Department of Statistics , Columbia University, New York, USA

## Abstract

The goal of causal discovery is to learn a directed acyclic graph from data. One of the most well-known methods for this problem is Greedy Equivalence Search (GES). GES searches for the graph by incrementally and greedily adding or removing edges to maximize a model selection criterion. It has strong theoretical guarantees on infinite data but can fail in practice on finite data. In this paper, we first identify some of the causes of GES's failure, finding that it can get blocked in local optima, especially in denser graphs. We then propose eXtremely Greedy Equivalent Search (XGES), which involves a new heuristic to improve the search strategy of GES while retaining its theoretical guarantees. In particular, XGES favors deleting edges early in the search over inserting edges, which reduces the possibility of the search ending in local optima. A further contribution of this work is an efficient algorithmic formulation of XGES (and GES). We benchmark XGES on simulated datasets with known ground truth. We find that XGES consistently outperforms GES in recovering the correct graphs, and it is 10 times faster.

## 1 INTRODUCTION

In the problem of causal discovery, we observe a multivariate data set $x^{1:n}$, where each $x^i = (x_1^i, ..., x_d^i)$. Our goal is to learn a $d$-node directed graphical model for $p(x_1^i, \ldots, x_d^i)$, i.e., a factorization of the joint distribution. In practice, causal discovery learns an equivalence class of graphs, called a *Markov equivalence class*, where each graph in the class implies the same set of conditional independence statements. The goal is to find the class whose set of independence statements exactly holds in the data.

The challenge to causal discovery is that the space of graphs

on $d$ nodes is prohibitively large. To this end, researchers have explored a number of ideas, including developing efficient tests for conditional independence [Spirtes et al., 2000, Zhang et al., 2011], restricting the space of graphs to a smaller class [Bühlmann et al., 2014, Fang et al., 2023] , or searching efficiently the space of graphs. One of the most theoretically sound methods is *greedy equivalence search* (GES) [Chickering, 2002b]. GES posits a proper scoring function for the graph (relative to the data) and then greedily optimizes it by inserting and deleting edges.

In the limit of large data, GES enjoys theoretical guarantees of reaching the true graph. However, with finite data, GES can fail to find the solution. In particular, its performance decreases for graphs with non-trivial number of edges, e.g. more than two parents per node. And so we cannot apply GES to the kinds of large-scale problems that we regularly encounter in machine learning. To this end, researchers have proposed computationally efficient approximations [Ramsey et al., 2017] and continuous relaxations with gradient-based optimization [Zheng et al., 2018, Brouillard et al., 2020]. These methods can handle more variables and denser graphs, but they do not enjoy the same guarantees.

In this paper, we improve on GES in two ways. First, we empirically examine the failure modes of GES and then use this analysis to propose better heuristics to explore the space of DAGs. Second, we develop superefficient algorithms for implementing the low-level graph operations that GES requires. Put together, these innovations describe extreme GES (XGES), a new algorithm for causal discovery.

XGES is more reliable and scalable than GES, and without sacrificing its important theoretical guarantees. While GES's performance degrades as the density of edges increases, XGES's performance remains stable. We study XGES on a battery of simulations. We find that XGES outperforms GES and its variants in all scenarios, achieving significantly better accuracy and faster runtimes.

**Related Work.** Causal discovery encompasses a wide range of methods [Glymour et al., 2019]. Here, we focus

on score-based methods, which posit a proper scoring rule, and proceed to find the sets of graphs that maximize it. The Greedy Equivalence Search (GES) algorithm maximizes it using a greedy search strategy [Chickering, 2002b]. Extensions and variants of GES include OPS [Chickering, 2002b], GIES [Hauser and Bühlmann, 2012], GDS-EEV [Peters and Bühlmann, 2014], and ARGES [Nandy et al., 2018].

Fast-GES (fGES) is a more efficient implementation of GES[Ramsey et al., 2017]. But we find that it does not reproduce exactly GES's search strategy and hurts performance (see Section 5). Selective GES (SGES) guarantees a polynomial worst-case complexity but has limited speed improvement in practice [Chickering and Meek, 2015].

Other works improve GES by randomly perturbing the search [Alonso et al., 2018, Liu et al., 2023]. However, they are computationally expensive.

More recently, *differentiable causal discovery* methods have been proposed to maximize the score using gradient-based methods [Zheng et al., 2018, Brouillard et al., 2020, Nazaret et al., 2024]. These methods can model causal relations using neural networks, but unlike GES and XGES (proposed here), their optimization procedures have no theoretical guarantees of converging to the true graph.

## 2 CAUSAL DISCOVERY AND GES

We first review the causal discovery problem and the necessary details of the greedy equivalence search (GES) method.

### 2.1 CAUSAL GRAPHICAL MODELS

Causal discovery aims to identify cause-and-effect relationships between random variables $\{X_1, ..., X_d\}$. We reason about causal relationships using causal graphical models (CGM). A CGM has two components:

1. a directed acyclic graph (DAG), $G^* = (V, E)$, where a node $j \in V$ represents variable $X_j$ and an edge $(j, k) \in E$ denotes a direct causal link from $X_j$ to $X_k$,

2. conditional distributions $p(X_j \mid X_{\mathrm{Pa}_j^{G^*}})$, defining the distribution of $X_j$ given its causal parents $X_{\mathrm{Pa}_j^{G^*}}$.

The joint distribution of the variables $X_1, ..., X_d$ writes:

$$p^*(X) = \prod_{j \in V} p^*(X_j \mid X_{\mathrm{Pa}_j^{G^*}}). \qquad (1)$$

The goal of causal discovery is to recover the graph $G^*$ from the joint distribution $p^*$ or from samples drawn from $p^*$.

However, multiple CGMs with different graphs can generate the same $p^*$. Two important concepts address this difficulty: faithfulness and Markov equivalence [Spirtes et al., 2000].

**Faithfulness.** In Eq. (1), $G^*$ induces a factorization of $p^*$, which, in turn, induces independencies between variables: each $X_j$ is independent of its non-descendants given its parents $X_{\mathrm{Pa}_j^{G^*}}$ [Pearl, 1988]. Reciprocally, we say that a distribution $p^*$ and a graph $G^*$ are *faithful* if all the independencies in $p^*$ are exactly those implied by $G^*$ and no more.

For example, $H = (\{1, 2\}, \{1 \rightarrow 2\})$ and $q = q(X_1)q(X_2)$ form a valid CGM. But the independence $X_1 \perp\!\!\!\perp X_2$ present in $q$ is not suggested by $H$. Rather, $q$ is faithful to $H' = (\{1, 2\}, \varnothing)$, which has no superfluous edges like $1 \rightarrow 2$.

Limiting the search to faithful graphs reduces the possible CGMs that could have generated $p^*$. But this is not enough.

**Markov Equivalence.** Two distinct graphs can both be faithful to $p^*$ if they induce the same set of independencies on $p^*$. For example, $A \rightarrow B \rightarrow C$ and $A \leftarrow B \leftarrow C$ impose the same set of independencies, $\{A \perp\!\!\!\perp C \mid B\}$.

Graphs inducing the same independencies are called *Markov equivalent*, they form *Markov equivalence classes* (MEC). Since $G^*$ is identifiable only up to Markov equivalence, the task of causal discovery becomes finding the MEC of $G^*$.

**Remark 1.** *With more assumptions (e.g. about the form of $p^*$) or special data (e.g., interventions), other causal discovery methods focus on identifying the possible $G^*$ beyond Markov equivalence. See Glymour et al. [2019] for an excellent review. We focus on Markov equivalence classes.*

### 2.2 SCORE-BASED CAUSAL DISCOVERY

In this work, we assume to have $n$ iid samples, denoted $\mathbf{D} = \{(x_1^i, ..., x_d^i)\}_{i=1}^n$, from a distribution $p^*$ that is faithful to some $G^*$. We aim to recover the MEC of $G^*$ from $\mathbf{D}$.

Greedy Equivalence Search (GES) searches for the MEC of $G^*$ among all the possible MECs. It does so by searching for the MEC whose DAGs maximize a specific score function. It is a particular case of score-based causal discovery.

Score-based methods assign a score $S(G; \mathbf{D})$ to every possible DAG $G$ given the data $\mathbf{D}$. The score function $S$ is designed to be maximized by the true graph $G^*$. This turns causal discovery into an optimization problem.

$$G^* = \arg\max_G S(G; \mathbf{D}) \qquad (2)$$

Some scores have properties that are important for GES.

**Definition 1.** *A score $S$ is* score equivalent *if it assigns the same score to all the graphs in the same MEC.*

A score equivalent $S$ enables defining the score of a MEC as the score of any of its constituent graphs.

**Definition 2** (Local Consistency, [Chickering, 2002b])**.** *Let $\mathbf{D}$ contain $n$ iid samples from some $p^*$. Let $G$ be any DAG and $G'$ be a different DAG obtained by adding the edge $i \rightarrow j$ to $G$. A score $S$ is* locally consistent *if both hold:*

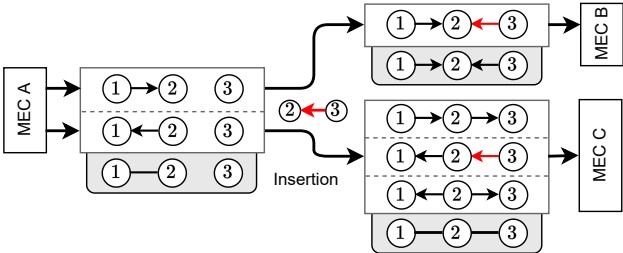

Figure 1: Illustration of insertions from a MEC A to MECs B or C: (i) choose a DAG in A, (ii) insert the edge $2 \leftarrow 3$ to obtain another DAG (iii) consider its MEC. Each MEC has all its DAGs on a white plate, and its canonical PDAG on a gray plate (see Section 4 for a definition).

1. $X_i \not\perp\!\!\!\perp_{p^*} X_j \mid X_{\mathrm{Pa}_j^G} \Rightarrow S(G'; \mathbf{D}) > S(G; \mathbf{D})$,

2. $X_i \perp\!\!\!\perp_{p^*} X_j \mid X_{\mathrm{Pa}_j^G} \Rightarrow S(G'; \mathbf{D}) < S(G; \mathbf{D})$.

*The independence statements are with respect to $p^*$.*

A locally consistent score increases when we add an edge that captures a dependency not yet represented in the graph. It decreases when we add an edge that does not capture any new dependency.

## 2.3 GREEDY EQUIVALENCE SEARCH

Chickering [2002b] introduced Greedy Equivalence Search (GES). It is a score-based method that is guaranteed to return the MEC of $G^*$ whenever the score is locally consistent. The main characteristic of GES is to navigate the space of MECs.

GES begins with the MEC of the empty DAG and iteratively modifies it to improve the score. At each step, only a few modifications are allowed. These modifications are of three types: insertions, deletions, and reversals.

**MEC modifications.** An *insertion* on MEC $M$ selects a DAG $G$ in $M$, adds an edge $x \to y$ to $G$ to obtain a different DAG $G'$ and replaces $M$ with the MEC of $G'$ (see Fig. 1).

A *deletion* on MEC $M$ selects a DAG $G$ in $M$, removes an edge $x \to y$ from $G$ to obtain a different DAG $G'$ and replaces $M$ with the MEC of $G'$.

A *reversal* on MEC $M$ selects a DAG $G$ in $M$, reverses an edge $x \to y$ to $x \leftarrow y$ in $G$ to obtain a different DAG $G'$ and replaces $M$ with the MEC of $G'$.

GES applies these modifications in three separate phases.

**Phase 1: Insert.** First, GES finds all possible insertions, applies the one leading to the largest score increase, and repeats until no insertion increases the score.

**Phase 2: Delete.** Then, GES finds all possible deletions, applies the one leading to the largest score increase, and repeats until no deletion increases the score.

---

**Algorithm 1:** Greedy Equivalence Search (GES)

**Input:** Data $\mathbf{D} \in \mathbb{R}^{n \times d}$, score function $S$
**Define:** $\delta_{\mathbf{D}, M}(O) = S(\mathrm{Apply}(O, M); \mathbf{D}) - S(M; \mathbf{D})$
**Output:** MEC of $G^*$

$M \leftarrow \{([d], \varnothing)\}$          // Empty graph's MEC
$\mathcal{I} \leftarrow$ get all insertions valid for $M$
**while** $|\mathcal{I}| > 0$ **do**
    $O^* \leftarrow \arg\max_{I \in \mathcal{I}} \{\delta_{\mathbf{D}, M}(I)\}$     // Get best insertion
    **if** $\delta_{\mathbf{D}, M}(O^*) \leq 0$ **then break**
    $M \leftarrow \mathrm{Apply}(O^*, M)$     // Apply best insertion
    $\mathcal{I} \leftarrow$ get all insertions valid for $M$

$\mathcal{D} \leftarrow$ get all deletions valid for $M$
**while** $|\mathcal{D}| > 0$ **do**
    $O^* \leftarrow \arg\max_{D \in \mathcal{D}} \{\delta_{\mathbf{D}, M}(D)\}$     // Get best deletion
    **if** $\delta_{\mathbf{D}, M}(O^*) \leq 0$ **then break**
    $M \leftarrow \mathrm{Apply}(O^*, M)$     // Apply best deletion
    $\mathcal{D} \leftarrow$ get all deletions valid for $M$

/* (Optional) 3rd phase like above but with reversals     */
**return** $M$

---

The MEC obtained at the end of phase 2 is exactly the MEC of $G^*$ if the score is locally consistent [Chickering, 2002b]

**Phase 3: Reverse.** In theory, phases 1 and 2 are sufficient to recover the MEC of $G^*$. Yet, Hauser and Bühlmann [2012] showed that adding a third phase with reversals can improve the search in practice with finite data (where the score might not be locally consistent). In this phase, GES finds all possible reversals for the MEC, applies the one that increases the score most, and repeats until none do.

The pseudocode of GES is given in Algorithm 1.

**Correctness.** GES's correctness relies on two properties: (i) the greedy scheme will reach the global maximum of any locally consistent score [Chickering, 2002b, Lemma 10], and (ii) the true graph $G^*$ and its MEC are the unique global maximizers of any locally consistent score [Chickering, 2002b, Proposition 8].

With these two properties, GES is guaranteed to recover the MEC of $G^*$ when the score is locally consistent.

**The BIC score.** A score commonly used by GES is the Bayesian Information Criterion (BIC) [Schwarz, 1978]. Given a model class $\mathcal{M}_G$ for each $G$, and our data $\mathbf{D}$ with its $n$ samples, the BIC defines a score:

$$S(G; \mathbf{D}) = \log p_{\hat{\theta}}(\mathbf{D}; G) - \frac{\alpha}{2} \log n \cdot |\hat{\theta}|, \qquad (3)$$

where $\alpha$ is a hyperparameter, $\hat{\theta}$ is the likelihood maximizer over $\mathcal{M}_G$, and the number of parameters $|\hat{\theta}|$ is usually the number of edges in $G$. The BIC is a model selection criterion trading off log-likelihood and model complexity. Models with higher BIC are preferred. The original BIC has $\alpha = 1$.

**Gaussian Linear Models.** A common model class used for

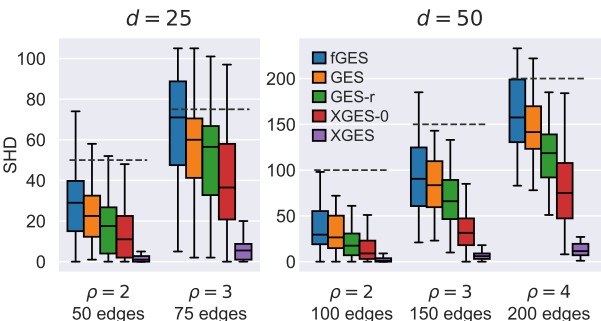

Figure 2: Performance comparison of GES and XGES variants, measured with SHD for different edge densities $\rho$. XGES heuristics outperform GES and its variants in all scenarios. The dashed lines indicate the number of edges of the true graph. Each boxplot is computed over 30 seeds.

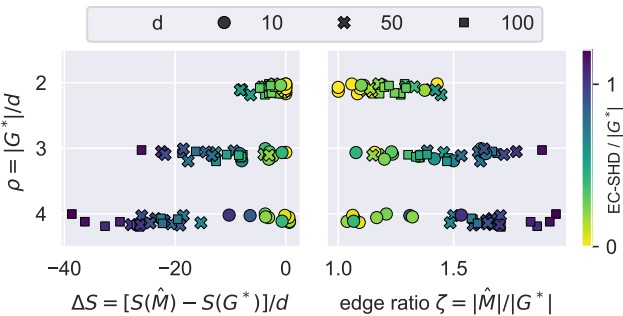

Figure 3: Empirical study of GES failure, on 90 simulated datasets with varying variables $d$ and graph densities $\rho$. (left) Differences in BIC between GES and ground-truth are negative. GES does not find the score's global maximum. (right) Ratios of GES-edges to true edges exceed 1. GES returns many more edges than the true graph.

continuous data with the BIC is the class of Gaussian linear models, where given a graph $G$, each variable is Gaussian with a linear conditional mean and a specific variance:

$$p_\theta(x_j \mid x_{\mathrm{Pa}_j^G}) \sim \mathcal{N}\Big(\sum_{k \in \mathrm{Pa}_j^G} \theta_{jk} x_k + \theta_{j0}, \theta_{j(d+1)}^2\Big). \quad (4)$$

For Gaussian linear models, the BIC is score equivalent. It can also be locally consistent under some conditions.

**Theorem 1** (Local Consistency of BIC [Haughton, 1988, Chickering, 2002b]). *For $\alpha > 0$, the BIC for Gaussian linear models is locally consistent once $n$ is large enough.*

Hence, for Gaussian linear models, GES is guaranteed to recover the MEC of $G^*$ with infinite data. In practice, however, data is finite and GES can return incorrect MECs.

**Example of Failure.** In Fig. 2, we report the performance of GES (orange) on simulated data, along with its variants and the proposed XGES. We simulate CGMs $(G^*, p^*)$ for $d \in \{25, 50\}$ variables, $\rho d \in \{2d, 3d, 4d\}$ edges ($\rho$ is an edge density parameter) and draw $n = 10,000$ samples from $p^*$. The simulation is detailed in Section 5.1. We compare the methods' results to the true graph $G^*$ using the structural Hamming distance for MECs (SHD), which counts the number of different edges between graphs of two MECs (see Section 5.1). Fig. 2 shows that GES can fail (SHD > 0), especially in denser graphs.

In addition, we confirm that including the third phase of reversals in GES improves its performance (GES-r, in green).

# 3 EXTREMELY GREEDY EQUIVALENCE SEARCH

In this section, we first investigate why GES can fail and then propose simple solutions to mitigate failure.

## 3.1 SCENARIOS OF GES FAILURE

As reviewed in Section 2.3, GES's correctness relies on two conditions: (i) the ability of its greedy search to find the score's global maximizer, and (ii) the true graph's MEC being the score's global maximizer (and the only one).

These two properties might not hold if the data cannot render $S$ locally consistent. We investigate GES failure. Is the issue that GES cannot reach the global maximizer with greedy search? If so, changing the search heuristic could help bypass local maxima. Or is it that GES effectively finds the global maximizer, but this maximizer is not the true graph? In this case, designing a new score might help. To check these hypotheses, we conduct empirical experiments.

**Greedy Optimization Fails.** We apply GES on the data simulated for Fig. 2, this time including $d \in \{10, 50, 100\}$ variables. We obtain a MEC $\hat{M}$ that we compare to $G^*$ with:

$$\Delta S = \frac{S(\hat{M}; \mathbf{D}) - S(G^*; \mathbf{D})}{d}. \quad (5)$$

A negative $\Delta S$ indicates that GES failed to identify a global maximizer and that $G^*$ still scores higher than $\hat{M}$ (note that it does not imply $G^*$ is the global maximizer). A positive $\Delta S$ demonstrates that $G^*$ is no longer the global maximizer and that GES legitimately identified a MEC with high score.

In Fig. 3 (left), we consistently find $\Delta < 0$ across different numbers of variables $d$ and edge densities $\rho$. This is evidence that GES fails to reach the global maximum and that the true graph $G^*$ still has a better score than $\hat{M}$.

We measure next the ratio $\zeta = |\hat{M}|/|G^*|$ between the number of edges found by GES and the number of true edges. Fig. 3 (right) shows that GES over-inserts edges i.e. $\zeta > 1$, especially with dense graphs. The idea behind GES is to over-insert edges in the first phase and then delete them in the second phase. However, we hypothesize that over-

**Algorithm 2:** XGES-0.

---

**Input:** Data $\mathbf{D} \in \mathbb{R}^{n \times d}$, score $S$.
**Define:** $\delta_{\mathbf{D},M}(O) = S(\text{Apply}(O, M); \mathbf{D}) - S(M; \mathbf{D})$
**Output:** MEC of $G^*$

$M \leftarrow \{([d], \varnothing)\}$
$\mathcal{I}, \mathcal{D}, \mathcal{R} \leftarrow$ all insertions, deletions, reversals valid for $M$
**while** $|\mathcal{I}| + |\mathcal{D}| + |\mathcal{R}| > 0$ **do**
    **if** $|\mathcal{D}| > 0$ **and** $\max_{D \in \mathcal{D}}\{\delta_{\mathbf{D},M}(D)\} \geq 0$ **then**
        $O^* \leftarrow \arg\max_{D \in \mathcal{D}}\{\delta_{\mathbf{D},M}(D)\}$
    **else if** $|\mathcal{R}| > 0$ **and** $\max_{R \in \mathcal{R}}\{\delta_{\mathbf{D},M}(R)\} > 0$ **then**
        $O^* \leftarrow \arg\max_{R \in \mathcal{R}}\{\delta_{\mathbf{D},M}(R)\}$
    **else if** $|\mathcal{I}| > 0$ **and** $\max_{I \in \mathcal{I}}\{\delta_{\mathbf{D},M}(I)\} > 0$ **then**
        $O^* \leftarrow \arg\max_{I \in \mathcal{I}}\{\delta_{\mathbf{D},M}(I)\}$
    **else**
        **break**       // No more operations available
    $M \leftarrow \text{Apply}(O^*, M)$
    $\mathcal{I}, \mathcal{D}, \mathcal{R} \leftarrow$ all insertions, deletions, reversals valid for $M$

**return** $M$

---

inserting may lead GES into local maxima before the second phase can correct it.

Following these two observations, we focus on ensuring that GES reaches the global maximizer. To do so, we design novel search heuristics aimed at preventing over-insertion.

## 3.2 THE HEURISTIC XGES-0

GES considers insertions, deletions, and optionally reversals in three separate phases. Rather, we consider all operations simultaneously, where deletions, insertions, and reversals can interleave in any order. And when both insertions and deletions can increase the score, we prioritize deletions.

**Heuristic XGES-0.** At each step, identify all the valid insertions, deletions, and reversals. If some deletes would increase the score, apply the best one. Otherwise, if some reversals would increase the score, apply the best one. Otherwise, apply the best insert. Repeat until no deletions, reversals, or insertions can increase the score.

We call this heuristic XGES-0 for eXtremely Greedy Equivalence Search and we detail it in Algorithm 2. XGES-0 retains the same theoretical correctness as GES.

**Theorem 2.** *For any locally consistent score $S$, the MEC $\hat{M}$ returned by XGES-0 contains the true graph $G^*$.*

The proof leverages the same theorems as those used to prove GES's correctness. It is provided in Appendix B.3.1.

In Fig. 2, we find empirically that XGES-0 (red) obtains MECs with better scores and closer to $G^*$ than GES. Early deletions effectively reduce encounters with local maxima.

**Remark 2.** *Alongside GES, Chickering [2002b] proposed*

*a variant called OPS that also considered insertions and deletions simultaneously. But OPS did not prioritize deletions over insertions, resulting in no improvements to GES in practice. We provide more details in Appendix A.5.*

## 3.3 THE HEURISTIC XGES

Building upon XGES-0, we introduce the heuristic XGES. XGES complements XGES-0. It repeatedly uses XGES-0, each time deleting an edge that causes a local maximum.

**Heuristic XGES.** XGES begins by applying XGES-0 until no operations can increase the score. Then, it enumerates all valid deletions, all of which will decrease the score. For each deletion, XGES copies the MEC and resumes XGES-0 on the copy until termination. But without ever reinserting the edge removed by the deletion. If the final score is worse than the original MEC, the copy is discarded, and the search continues with the next deletion. If the final score is better, the copy becomes the new MEC. XGES then restarts with the new MEC and all its new deletions. XGES stops once all deletions of a MEC have been tried. We provide the pseudocode of XGES in Algorithm 3 in Appendix B.3.2.

**Intuition.** The XGES heuristic aims to remove incorrect edges that were inserted early in the search and might be causing local maxima preventing their deletion. By forcefully deleting these edges, XGES can get around the local maximum and discover better graphs.

**Theorem 3.** *For any score $S$, XGES returns a MEC $\hat{M}$ with a higher or equal score than XGES-0. If $S$ is locally consistent, then $\hat{M}$ contains the true graph $G^*$.*

The proof follows from the design of XGES and by Th. 2.

Fig. 2 illustrate the performance of XGES (purple), showing that it significantly improves over all other GES variants. XGES enables non-trivial causal discovery in denser graphs ($\rho \geq 2$) with many variables ($d \geq 50$). However, XGES is computationally more expensive than XGES-0. To alleviate this, we develop an efficient implementation for it.

## 4 EFFICIENT ALGORITHM

GES, if naively implemented, is slow. In this section, we develop new ways of implementing its details to significantly speed it up. As we will see in the empirical studies, these details are crucial to scaling up XGES to large dense graphs. We now review how GES manipulates MECs in practice, and then show how to make it more efficient.

## 4.1 MANIPULATING MECS WITH CPDAGS

MECs are sets of DAGs whose size can grow exponentially with the number of nodes $d$ [He et al., 2015]. To manipulate them practically, GES builds on the following theorem.

**Theorem 4** ([Verma and Pearl, 1991]). *Two DAGs are Markov equivalent if and only if they have the same skeletons and the same v-structures.*

The *skeleton* of a graph is the undirected graph obtained by removing the direction of all edges; a *v-structure* is a triple of nodes such that $x \rightarrow y \leftarrow z$ with no edge between $x$ and $z$.

Th. 4 shows that all the graphs of a MEC share the same skeleton and differ only on edges that can be reversed without changing the set of v-structures. So within a MEC, some edges are consistently oriented in one direction while others may have different orientations between graphs. They are respectively called *compelled* and *reversible* edges.

**CPDAGs.** Each MEC can be represented by a *partially directed acyclic graph* (PDAG). A PDAG is a graph with both directed and undirected edges and no cycles of directed edges. The *canonical PDAG of a MEC* contains all the compelled edges as directed edges and all the reversible edges as undirected edges (see Fig. 1). A PDAG that is the canonical representation of a MEC is called a *completed PDAG* (CPDAG). A PDAG $P$ that is not a CPDAG but has the same skeleton and v-structures as another CPDAG $P'$ can be transformed into $P'$ with a method called *completing the PDAG* [Meek, 1995, Chickering, 2002a].

We will use the following terminology when discussing a PDAG. For node $x$: its *neighbors* $\mathrm{Ne}(x)$ are its neighbors from undirected edges, its *children* $\mathrm{Ch}(x)$ are its children from directed edges, its *parents* $\mathrm{Pa}(x)$ are its parents from directed edges, and its *adjacent* nodes $\mathrm{Ad}(x)$ are any of all three. A *semi-directed path* from $x$ to $y$ is a path from $x$ to $y$ with edges that are either undirected or directed toward the direction of $y$. A *clique* is a set of all adjacent nodes.

**Operators on CPDAGs.** GES associates each operation on a MEC $M$ with an operator acting on its CPDAG $P$, such that an operation changing $M$ into $M'$ is associated with an operator changing $P$ into $P'$, the CPDAG of $M'$.

For insertions, the operators used by GES are of the form $\mathrm{Insert}(x, y, T)$ where $x, y \in V$ and $T \subset V$. The action of $\mathrm{Insert}(x, y, T)$ on $P$ is to insert the edge $x \rightarrow y$, orient any undirected edges $t - y$ as $t \rightarrow y$ for $t \in T$ and finally complete the resulting PDAG into a CPDAG.

Given a MEC $M$ and its CPDAG $P$, Chickering [2002b] shows that there is a bijection between (a) the set of possible insertions on $M$ and (b) the set of operators $\mathrm{Insert}(x, y, T)$ satisfying the following validity conditions relative to $P$:

**I1.** $x \notin \mathrm{Ad}(y)$. (6)

**I2.** $T \subset \mathrm{Ne}(y) \setminus \mathrm{Ad}(x)$. (7)

**I3.** $(\mathrm{Ne}(y) \cap \mathrm{Ad}(x)) \cup T$ is a clique. (8)

**I4.** All semi-directed paths from $y$ to $x$ have a node in
$(\mathrm{Ne}(y) \cap \mathrm{Ad}(x)) \cup T$. (9)

Insert operators satisfying these conditions, with $\mathrm{Ad}, \mathrm{Ne}$,

Pa, clique and paths computed in $P$, are called *valid* for $P$. To navigate from one MEC to another with an insertion, GES applies the corresponding valid Insert operator from one CPDAG to another.

**Score of Operators.** The increase in score after an Insert operation can be efficiently computed when the score is BIC. Indeed, the BIC for a graph $G$ equivalently rewrites as:

$$S(G; \mathbf{D}) = \sum_{j=1}^{d} s(j, \mathrm{Pa}_j^G; \mathbf{D}), \qquad (10)$$

where $s(j, \mathrm{Pa}_j^G; \mathbf{D})$ is called the *local score* of $j$ and equals:

$$\sum_{i=1}^{n} \log p_{\hat{\theta}}(x_j^i | x_{\mathrm{Pa}_j^G}^i) - \frac{\alpha}{2} \log n \cdot | \mathrm{Pa}_j^G |. \qquad (11)$$

A score decomposing as Eq. (10) is called *decomposable*.

With a decomposable score, the increase in score for an operator $\mathrm{Insert}(x, y, T)$ applied to $P$ is:

$$\delta = s(y, (\mathrm{Ne}(y) \cap \mathrm{Ad}(x)) \cup T \cup \mathrm{Pa}(y) \cup \{x\})$$
$$- s(y, (\mathrm{Ne}(y) \cap \mathrm{Ad}(x)) \cup \mathrm{Pa}(y)), \quad (12)$$

where each term $\mathrm{Ad}, \mathrm{Ne}, \mathrm{Pa}$ is computed relative to $P$. For convenience, we say that $\delta$ is the *score* of the operator.

Similar derivations are made for Delete and Reversal in Chickering [2002b] and Hauser and Bühlmann [2012] (reversal is called turning). We review them in Appendix B.4.1.

**GES with CPDAGs.** In sum, GES implements Algorithm 1 using CPDAGs and operators. It begins with the empty CPDAG, identifies all the Insert (or Delete, Reversal) that are valid for the current CPDAG, computes their scores, applies the best one if it has a positive score, and repeats.

XGES could proceed similarly. However, whether for GES or XGES, constructing the list of valid operators and scoring them at each step is computationally expensive. We now turn to new ways to more efficiently implement these operations.

## 4.2 EFFICIENT ALGORITHMIC FORMULATION

When applying an operator on $P$ to form $P'$, the validity conditions of the other operators (Eqs. (6) to (9)) can become valid or invalid. Similarly, the score of the other operators in Eq. (12) can change. Yet, as noticed in Ramsey et al. [2017], only a few edges changed from $P$ to $P'$. As a result, most other operators that were computed for $P$ but not applied remain valid operators for $P'$. Similarly, the scores of most operators remain identical.

Each step of XGES involves the following sub-steps:

1. Start with a CPDAG $P$ and a list of candidate operators $\mathcal{C}$, where $\mathcal{C}$ is guaranteed to include all the valid operators for $P$, and their scores.

| | Pre-update | $a$ $b$ | $a \rightarrow b$ |
|---|---|---|---|
| | Post-update | $a - b$ | $a - b$ |
| Necessary conditions | | $y \in \{a, b\}$ or $y \in \mathrm{Ne}(a) \cap \mathrm{Ne}(b)$ or $(x = a) \wedge (y \in \mathrm{Ne}(b))$ or $(x = b) \wedge (y \in \mathrm{Ne}(a))$ | $y \in \{a, b\}$ |

Table 1: Necessary conditions for an Insert$(x, y, T, E)$ to become valid after the $(a, b)$ update. Excerpt of Table 4 from Appendix B.5 with only two types of updates.

2. Choose the best operator $O^*$ from $\mathcal{C}$ using XGES's heuristic (deletion before reversal, before insertion).

3. Verify that $O^*$ is valid for $P$, otherwise re-run the heuristic on $\mathcal{C} \setminus \{O^*\}$ until a valid operator is found.

4. Apply $O^*$ to $P$ to form $P'$ and add to $\mathcal{C}$ all the operators that became valid for $P'$, with their scores. Return to 1, with $P \leftarrow P'$, as we have just guaranteed that $\mathcal{C}$ includes all the valid operators for $P$.

The operators that became invalid for $P'$ are not removed from $\mathcal{C}$. It is more efficient to leave them in the list and only check the validity of an operator in step 3 just before applying it (and discarding it if invalid). Indeed, if we recheck the validity of all operators after each operation, a single operator will be rechecked at each step until it is applied, instead of being checked only once before being applied.

No steps were included to recompute the scores of any operators in $\mathcal{C}$. We explain how we can avoid it next.

### 4.2.1 Updating the Score of Operators.

To avoid recomputing the scores of operators at each step, we change the parametrization of the operators to make their scores independent of the CPDAG they are applied to.

We parametrize each Insert by an additional set $E \subset V$ and an extra validity condition that completes Eqs. (6) to (9):

**I5**.  $\quad E = (\mathrm{Ne}(y) \cap \mathrm{Ad}(x)) \cup T \cup \mathrm{Pa}(y).$  (13)

The score of Insert$(x, y, T, E)$ from Eq. (12) becomes $s(y, E \cup \{x\}) - s(y, E)$, which only depends on the Insert parameters. We reparametrize Delete and Reversal operators similarly in Appendix B.4.2.

With Chickering [2002b]'s parametrization, the score of an Insert would change if $(\mathrm{Ne}(y) \cap \mathrm{Ad}(x)) \cup T \cup \mathrm{Pa}(y)$ changes. Now, with $E$ as a fixed parameter of the operator, it is the status of condition **I5** that would change. We turned a change in score into a change in validity.

We now turn to efficiently update the valid operators.

### 4.2.2 Updating the Validity of Operators.

After updating a CPDAG $P$ into $P'$, our goal is to efficiently add to $\mathcal{C}$ the operators that became valid for $P'$.

To do so, we decompose the update from $P$ to $P'$ into a succession of single edge updates $P_1, \ldots P_k$, with $P_1 = P$, $P_k = P'$ and where $P_i$ and $P_{i+1}$ only differ on the orientation or presence of a single edge, e.g. $a \rightarrow b$ vs $a - b$. We then have the following theorem.

**Theorem 5.** *Write $P_1, \ldots P_k$ a sequence of single edge updates that transforms $P$ into $P'$. Take an operator $O$ that is invalid for $P$ and becomes valid for $P'$ and write $\{c_1, \ldots, c_m\}$ its validity conditions, e.g. **I1** to **I5** for an Insert. Then there exists $i^* \in \{1, k-1\}$ and one validity condition $c_{j^*}$ such that $c_{j^*}$ is false for $P_{i^*}$, true for $P_{i^*+1}$, and all other conditions $c_j \neq c_{j^*}$ are true for $P_{i^*+1}$.*

*Proof.* All $c_j$ are true for $P'$ i.e. $P_k$. So let us step back from $P'$ to $P$ until one of the conditions $c_{j^*}$ becomes false for some $P_{i^*}$. Such an $i^*$ must exist since some condition is false for $P$ i.e. $P_1$. $P_{i^*}$ and $c_{j^*}$ satisfy the theorem. $\quad \square$

With Th. 5, we can efficiently update $\mathcal{C}$ if we can identify which operators are susceptible to having one of their conditions become true after single-edge updates.

In Appendix B.5 we study the necessary conditions on the parameters of an operator to have one of its validity conditions become true after a single-edge update. We report the necessary conditions for all validity conditions of all operators against all types of edge updates in Table 4 in Appendix B.5. We provide an excerpt in Table 1 with only two types of edge updates, for the Insert operator only, and where we grouped the necessary conditions for each validity condition into a single set of necessary conditions (with or).

For example, if edge $a \rightarrow b$ is changed into $a - b$, Table 1 shows that the only Insert$(x, y, T, E)$ that can become valid are those with $y \in \{a, b\}$. If the edge $a - b$ is changed into $a \rightarrow b$, then the necessary conditions for an Insert operator to become valid are more involved but still efficient.

In sum, we can efficiently update $\mathcal{C}$ after each CPDAG update using Table 4 in Appendix B.5.

### 4.2.3 XGES Implementation.

We implement the efficient algorithmic formulation of XGES-0 and XGES at `github.com/ANazaret/XGES`. We provide code in C++ as well as a Python wrapper.

## 5 EMPIRICAL STUDIES

We compare the XGES heuristics to different variants of GES. We find that XGES recovers causal graphs with sig-

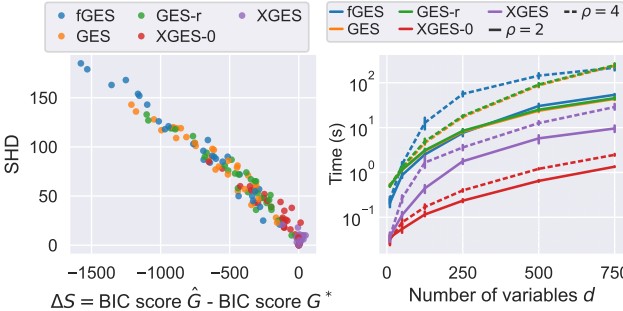

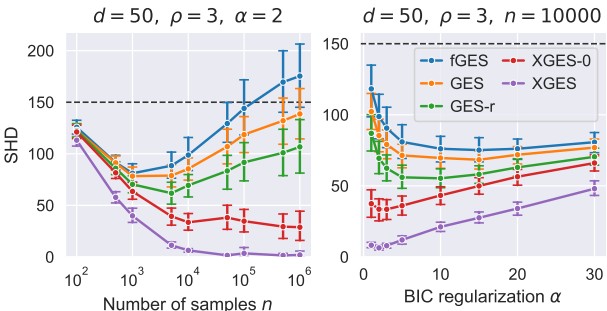

Figure 4: (left) The BIC scores of the graphs returned by each method are strongly correlated with the SHD to ground truth (shown for $d = 50$, $\rho = 3$, 30 seeds). XGES finds the highest scores and lowest SHDs. (right) Runtime of GES and XGES for a wide range of $d$. XGES-0 is up to 30 times faster than GES, and XGES up to 10 times faster. fGES may have overhead due to Java while other methods are in C++.

Figure 5: Performance of GES and XGES when varying (left) the number of samples $n$, and (right) the regularization strength $\alpha$. Increasing $n$ improves XGES while it hurts GES and its variants. Increasing $\alpha$ initially improves GES but eventually hurts all methods. The dashed lines indicate the number of edges of the true graph. Error bars over 30 seeds.

nificantly better accuracy and up to 10 times faster.

## 5.1 EVALUATION SETUP

**Data Simulation.** We simulate CGMs and data for different numbers of variables $d$, edge density $\rho$ (average number of parents) and number of samples $n$. We first draw a random DAG $G^*$ from an Erdos-Renyi distribution. We then obtain $p^*$ by choosing each conditional distribution $p^*(x_i \mid x_{\text{Pa}_{G^*}^i})$ as a Gaussian $x_i \sim \mathcal{N}(W_i^\top x_{\text{Pa}_{G^*}^i}, \varepsilon_i)$ where $W_i, \varepsilon_i$ are random selected. To ensure faithfulness, we sample $W_i$ away from 0. More details are in Appendix A.7.1.

**Baseline Algorithms.** We compare our algorithms against GES without reversals (GES), and with reversals (GES-r, a.k.a GIES), using the C++ implementation in the R package `pcalg` [Kalisch et al., 2012]. We also include fast-GES (fGES) from the Java software Tetrad [Ramsey et al., 2017]. An additional baseline, OPS, is provided in Appendix A.5.

**Evaluation Metrics** We evaluate the algorithms with the structural Hamming distance on MECs (SHD) between the method's results $\hat{M}$ and the ground-truth MEC $M^*$ [Peters et al., 2014]. The SHD is the number of different edges between the CPDAGs of $\hat{M}$ and $M^*$. We also consider causal discovery as a binary classification task, where $\hat{M}$ predicts the presence of edges in $M^*$. We report the F1 score, precision, and recall for this task. Error bars are computed over multiple random datasets (seeds) and reported as bootstrapped 95% confidence intervals [Waskom, 2021].

## 5.2 RESULTS

**General Performance.** In Fig. 2, we find that XGES-0 and XGES outperform all baselines. The improvement is more significant for larger density $\rho$ and larger $d$. The con-

clusions are identical with precision and recall in Fig. 7 of Appendix A.2, which are both improved by XGES. We emphasize that even though XGES favors deleting edges, the proportion of true edges recovered by XGES is higher than GES (the recall). We also report the F1 metric in Appendix A.2. We note that the performance of fGES is slightly worse than GES. We explain in Appendix B.6 that one of the optimizations of fGES removes some valid insertions.

**Choice of Metrics.** Fig. 4 (left) shows that the BIC scores of the graphs returned by each method are strongly correlated with the SHD to ground truth. This is a comforting observation: maximizing the BIC score on finite data is indeed a good proxy for minimizing SHD to ground-truth.

**Impact of the Edge Density $\rho$.** In Fig. 9, we see that when $\rho = 1$ (a very sparse graph where nodes have $\rho = 1$ parent on average), then all methods perform similarly well. The advantage of XGES over GES is visible as soon as $\rho = 2$ and widens as $\rho$ increases, see also Fig. 2.

**Robustness to the Sample Size $n$.** In Fig. 5 (left), we vary the number of samples $n$ and fix $d = 50$ and $\rho = 3$. XGES's performance improves with $n$, coherent with Th. 1. In contrast, GES and its variants are hurt when $n$ increases beyond $10^4$. But this is not incoherent with GES's correctness in the limit of infinite data. Instead, this reveals that the finite sample behavior of GES is nontrivial and that GES may require very large $n$ – beyond what is practical – to perform well.

In Fig. 12, we study sample sizes up to $n = 10^8$ on a small graph with $d = 15$ and $\rho = 2$. We find again that GES worsens around $10^4$ samples, but this time, it improves again after $10^5$ samples, thereby exhibiting a double descent behavior. We discuss it in more detail in Appendix A.6.

**Robustness to the Regularization Strength $\alpha$.** In Fig. 5 (right), we vary $\alpha$ and fix $d = 50$, $\rho = 3$ and $n = 10000$. We find that increasing $\alpha$ helps GES from $\alpha = 1$ to $\alpha = 10$

but then hurts it. No value of $\alpha$ enables GES to catch up to XGES. Echoing Section 3.1, we conclude that the solution to GES's over-inserting is not to change the score function, but to change the search strategy, as XGES does.

**Robustness to the Data Simulation.** We vary the procedure to sample the weights $W_i$ in two ways: changing the scale and changing the shape of their distribution. We report the results in Appendix A.7 with Figs. 13 and 14. We find similar conclusions as in Fig. 2.

**Implementation Speed.** We measure the runtime of the different methods for a wide range of $d$ and $\rho \in \{2, 4\}$ in Fig. 4 (right). We find that XGES-0 is up to 30 times faster than GES, and XGES up to 10 times faster. While fGES's slower runtime may be attributed to Java overhead, the other methods are implemented in C++. Higher densities slow down all methods, with a stronger impact on GES, which is coherent with GES over-inserting in denser graphs.

We find the same conclusions by reporting the number of calls to the scoring function as another measure of efficiency in Appendix A.4. Interestingly, even though XGES repeatedly applies XGES-0, it only makes around one order of magnitude more BIC score evaluations than XGES-0.

# CONCLUSION AND FUTURE WORK

We introduced XGES, an algorithm that significantly improves on GES. With XGES, we can learn larger and denser graphs from data. XGES offers several avenues for future work. One direction is to study its finite sample guarantees. A second is to study its applicability to more complex model classes beyond linear models. Another is to relax its assumptions: e.g. unfaithful graphs, or scores that are not asymptotically locally consistent [Schultheiss and Bühlmann, 2023]. Finally, its efficient implementation could be used to analyze large real-world datasets.

### Acknowledgements

We thank the anonymous reviewers for their helpful comments. A.N. was supported by funding from the Eric and Wendy Schmidt Center at the Broad Institute of MIT and Harvard, and the Africk Family Fund. D.B. was funded by NSF IIS-2127869, NSF DMS-2311108, NSF/DoD PHY-2229929, ONR N00014-17-1-2131, ONR N00014-15-1-2209, the Simons Foundation, and Open Philanthropy.

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

# Extremely Greedy Equivalence Search
## (Supplementary Material)

**Achille Nazaret**[1]  **David Blei**[1,2]

[1]Department of Computer Science, Columbia University, New York, USA
[2]Department of Statistics , Columbia University, New York, USA

## A EMPIRICAL STUDIES

### A.1 SAMPLE SIZE AND REGULARIZATION STRENGTH

We reproduce figure Fig. 5 from the main text in Fig. 6a, which shows the impact of the sample size $n$ and the regularization strength $\alpha$ on the performance of GES and XGES variants. We complete it with Fig. 6b to show the corresponding number of predicted edges. It indeed reveals that increasing $n$ causes GES and its variants to over-insert.

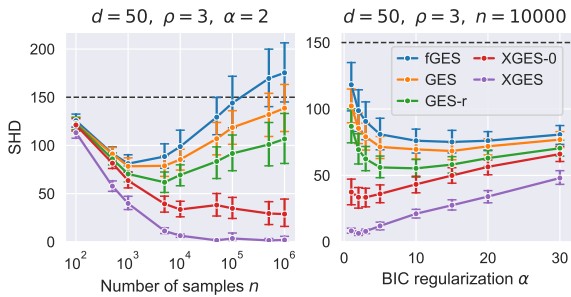

(a) Evolution of the performance of GES and XGES variants with the number of samples $n$.

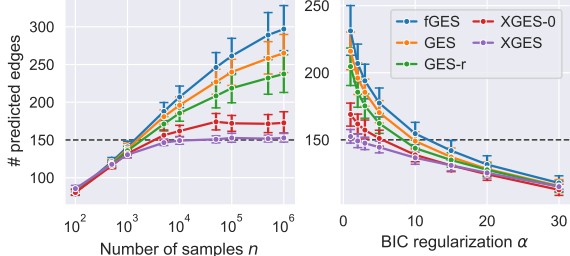

(b) Evolution of the number of predicted edges with the number of samples $n$.

Figure 6: Evolution of the performance of GES and XGES variants with (left) the number of samples $n$, and (right) the regularization strength $\alpha$. Increasing $n$ hurts GES and its variants (as it reduces the BIC regularization and over-insertion is exacerbated) while it helps XGES. Increasing the BIC regularization with $\alpha$ helps GES but without letting it catch up with XGES. Missing points indicate the method did not run.

### A.2 COMPLEMENTARY METRICS: PRECISION, RECALL, F1 SCORE, BIC SCORE.

We complement Fig. 2 from the main text with Fig. 7 to show the F1 score of GES and XGES for the same simulated data, as well as the precision/recall breakdown. We also report the BIC score of the graphs returned by GES and XGES in Fig. 8. We find similar conclusions as in the main text.

The methods return a MEC $\hat{M}$ to predict the true MEC $M^*$. The F1 score, precision and recall are defined for binary classification problems. For each ordered pair of nodes $(i, j)$ we say $M$ contains $(i, j)$ if $(i, j)$ is directed in $M$ from $i$ to $j$ or if $(i, j)$ is undirected in $M$ $(i, j)$ (but not if $(j, i)$ is directed in $M$ from $j$ to $i$). Then, the binary classification problem is to predict for each $(i, j)$ if $M^*$ contains $(i, j)$ or not, predicted by whether $\hat{M}$ contains $(i, j)$ or not.

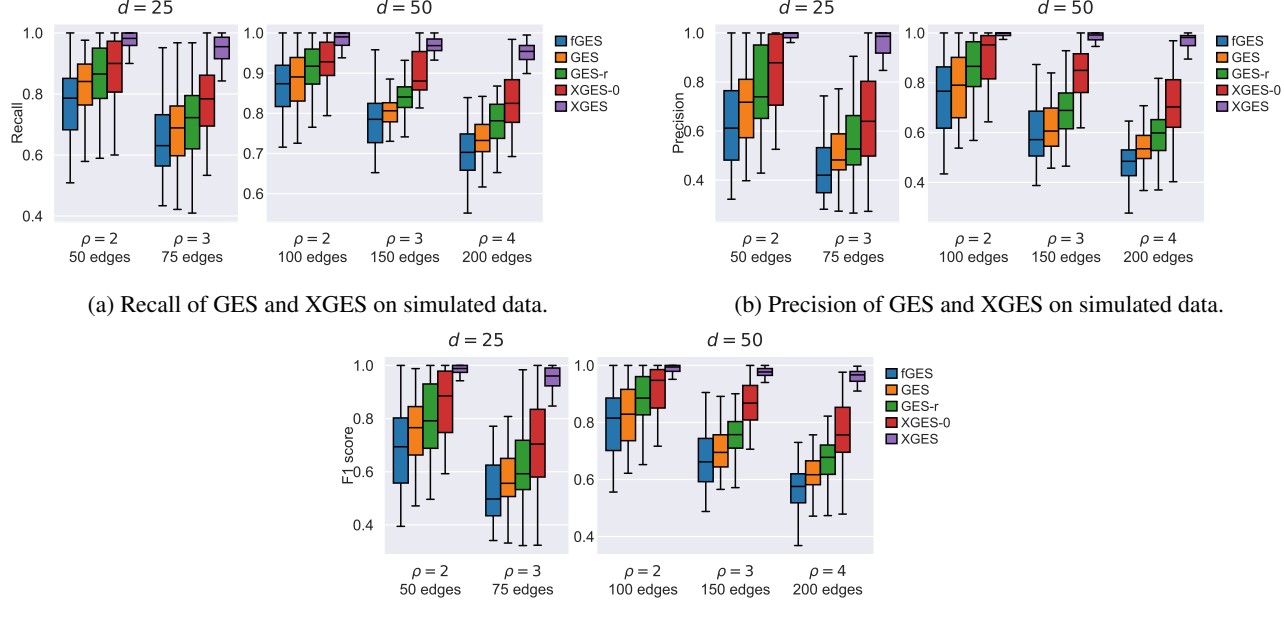

(a) Recall of GES and XGES on simulated data.

(b) Precision of GES and XGES on simulated data.

(c) F1 of GES and XGES on simulated data.

Figure 7: Performance of GES and XGES on simulated data measured with precision, recall and F1. With $n = 10000, \alpha = 2$ and 30 seeds (same data as Fig. 2). XGES outperforms other methods in the three metrics.

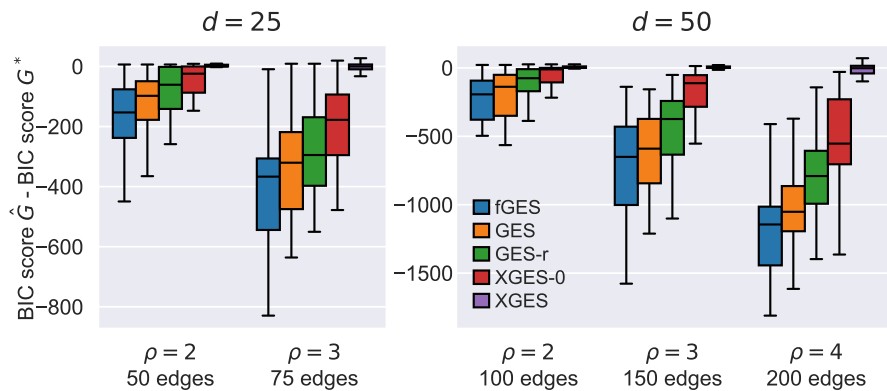

Figure 8: Difference in score between the true graph/MEC $G^*$ and the graph/MEC returned by GES and XGES on simulated data. XGES returns graphs with higher scores than GES.

## A.3 MORE VARIABLES AND EDGE DENSITIES

We show the performances of the methods on more combinations of $d$ and $\rho$ in Fig. 9. We fixed $n = 10000, \alpha = 2$. We find similar trends as in Fig. 2 and Fig. 4.

For edge density $\rho = 1$, we find no significant difference between GES and XGES. This is expected as the true graph is really sparse and the methods are less likely to encounter local optima.

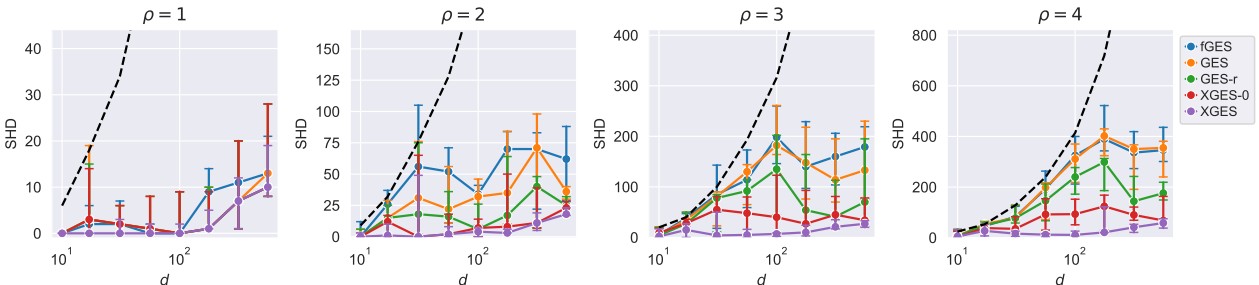

Figure 9: Performance of GES and XGES on more combinations of $d$ and $\rho$. We fixed $n = 10000, \alpha = 2$ and we report error bars and averages over 5 seeds. XGES consistently outperforms GES and its variants. The dashed lines indicate the number of true edges.

## A.4 NUMBER OF CALLS TO THE SCORING FUNCTION

We show the number of calls to the scoring function for the different methods in Fig. 10. We find that even though XGES repeatedly applies XGES-0, it only makes around one order of magnitude more BIC score evaluations. fGES was not included because we could not extract the number of calls from the Tetrad implementation.

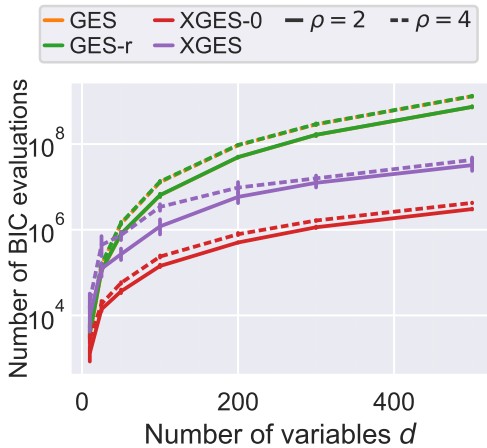

Figure 10: Number of calls to the scoring function for the different methods. We fixed $n = 10000, \alpha = 2$, and we report error bars and averages over 5 seeds.

## A.5 THE OPS VARIANT

Chickering [2002b] evaluated GES against a variant called OPS that also considered insertions and deletions simultaneously. But OPS did not prioritize deletions over insertions, resulting in limited changes to GES in their experiments. We show the performance of OPS in Fig. 11. We corroborate the results of Chickering [2002b] that OPS has performances similar to GES.

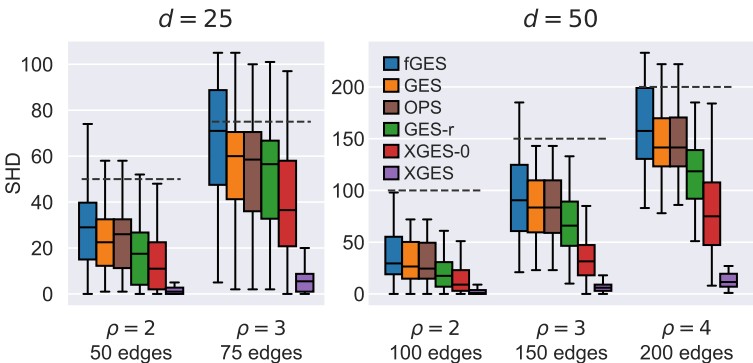

Figure 11: Same experiment as Fig. 2 but with the addition of the OPS variant. OPS performs very similarly to GES, suggesting that XGES-0's heuristic favoring deletions over insertions in XGES-0 is important.

Mathematically, deleting an edge can only increase the BIC score by at most $\frac{\alpha}{2} \log n$ which is usually smaller than the increase from inserting an edge. Hence even if OPS considers deletions and insertions "together", we conjecture that most deletions are only considered at the end of the search because insertions have higher scores and are applied first. OPS then encounters the same local maxima as GES. This highlights the importance of prioritizing deletions over insertions with XGES.

### A.6 THE DOUBLE DESCENT OF GES WITH THE SAMPLE SIZE

We show the performance of GES and XGES on a small graph with $d = 15$ and $\rho = 2$ for sample sizes up to $n = 10^8$ in Fig. 12. XGES monotonically and quickly improves to an SHD of 0. We find that GES improves from $n = 10^2$ to $n = 10^4$ but worsens around $n = 10^4$. It eventually improves again after $10^6$. The error bands are bootstrapped 95% confidence intervals over 30 seeds. We chose the graph to be small so that we could observe the second descent of GES with sample sizes up to $10^8$. However, we doubt that such large sample sizes are practical. We further believe that the issue worsens with larger graphs. (*Note: We reimplemented GES and GES-r to scale to $n = 10^8$, we verified that we obtained the same results as the original implementations for $n$ up to $10^6$. See parameter $-b$ in the XGES code.*)

When the sample size increases, the strength of the BIC regularization relative to the likelihood decreases in $\frac{\log(n)}{n}$. We hypothesize that GES worsens because that decrease might lead to over-inserting, and the encounter of local optima.

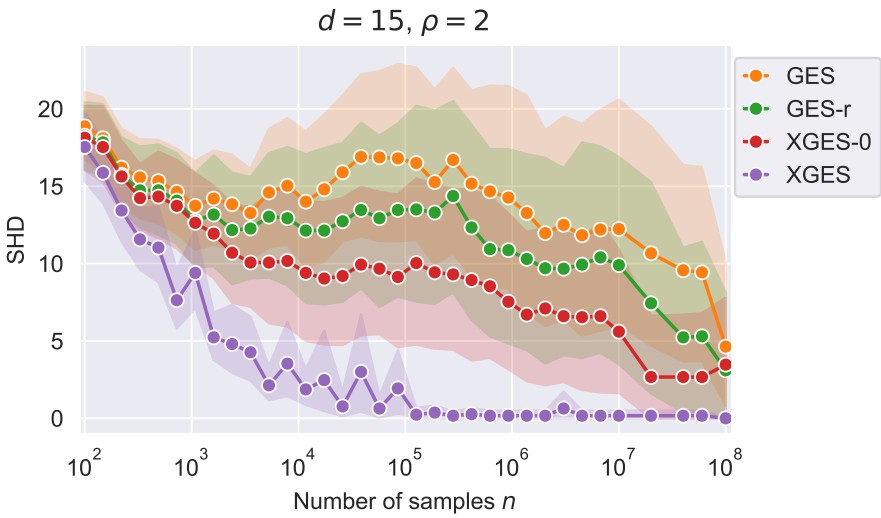

Figure 12: Performance of GES and XGES on a small graph with $d = 15$ and $\rho = 2$ for sample sizes up to $n = 10^8$. We find that GES worsens around $n = 10^4$ but improves after $n = 10^6$. It exhibits a double descent behavior with the sample size.

## A.7 THE IMPACT OF THE SIMULATION

We describe the simulation procedure and show the impact of changing it.

### A.7.1 The simulation procedure

In the experiments, we use the following procedure to obtain simulated data.

1. Select $d$ and $\rho$.
2. Draw a DAG $G^*$ as follows:
   - $G^* \sim \text{Erdos-Renyi}(\#nodes = d, p = \frac{2\rho}{d-1})$.
   - Orient each edge $(i, j)$ from $\min(i, j)$ to $\max(i, j)$, that forms a DAG.
   - Draw a permutation $\sigma$ of $[d]$ and relabel each node $i \mapsto \sigma(i)$.
3. Choose a distribution $p^*$ as follows for each $i \in [d]$:
   - Sample weights $W_i$ away from 0 to ensure faithfulness.
     - (a) In most simulations, we draw $W_i \in [1, 3]^{\left|\text{Pa}_{G^*}^i\right|}$ from $\mathcal{U}([1, 3])$.
     - (b) In Fig. 14, we change the simulation to sample weights $W_i \in [-3, -1] \cup [1, 3]$, by additionaly drawing a Bernoulli variable $b_i \sim \mathcal{B}(0.5)$ and setting $W_i \leftarrow -W_i$ if $b_i = 1$.
   - $L_1$-normalize $W_i$ as $W_i \leftarrow W_i / \sum_j |W_{ij}|$.
   - Draw the scale of the Gaussian $\varepsilon_i \sim \mathcal{U}(0, \varepsilon_{\max})$.
     - (a) By default in most plot $\varepsilon_{\max} = 0.5$.
     - (b) We vary it in Fig. 13.
   - Define $p^*(x_i \mid x_{\text{Pa}_{G^*}^i}) = \mathcal{N}(W_i^\top x_{\text{Pa}_{G^*}^i}, \varepsilon_i^2)$.
4. Draw $n$ i.i.d. samples from $p^*$.

### A.7.2 Varying the scale of the Gaussian noise

We show the impact of varying the scale of the Gaussian noise $\varepsilon_{\max}$ in Fig. 13. As expected, GES and XGES are almost not impacted by the scale of the noise. This is expected, as the methods explicitly model the noise variable.

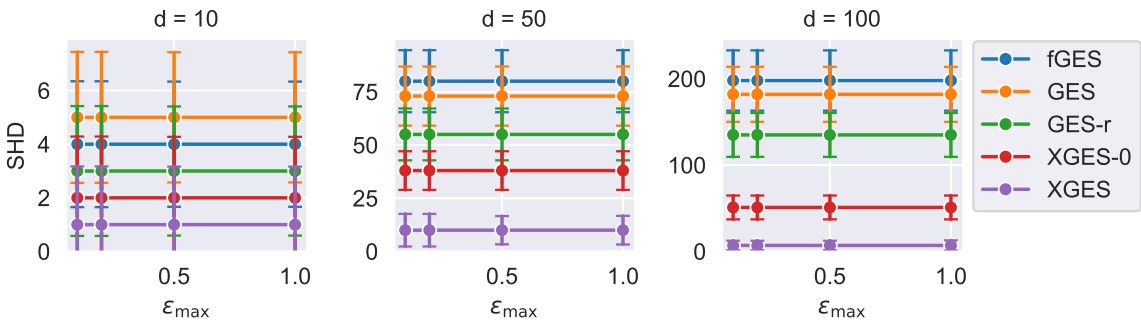

Figure 13: Performance of GES and XGES on different scales of the Gaussian noise. We fixed $n = 10000, \alpha = 2$.

### A.7.3   Different sampling of weights

We find similar conclusions when we change the sampling of weights from $W_i \in [1, 3]$ to $W_i \in [-3, -1] \cup [1, 3]$ . We show the results in Fig. 14.

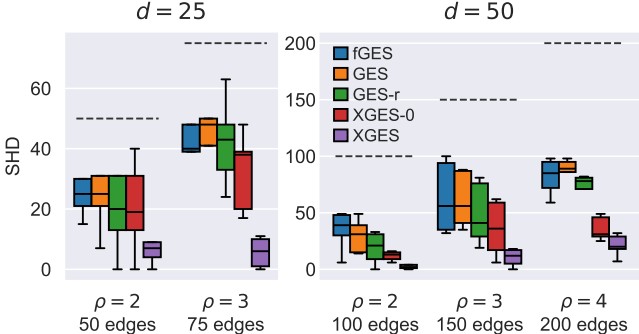

Figure 14: Performance of GES and XGES similar to Fig. 2 but with a different sampling of weights $W$. $n = 10000, \alpha = 2$.

# B THEORETICAL DETAILS

We detail the theoretical guarantees of GES and XGES. We first recall the problem only in terms of MECs and not CPDAGs. We review the theoretical guarantees of GES and prove those of XGES. We then show how the MEC formulation can be translated into a CPDAG formulation for Deletes and Reversals.

**Notations and Vocabulary.**

- $d$: number of variables.
- $G$: a DAG over $d$ variables.
- $P$: a PDAG over $d$ variables.
- $M$: a MEC over $d$ variables.
- $\mathcal{M}$: space of all MEC over $d$ variables.
- $S$: a locally consistent score that is score equivalent.
- $S(G)$ or $S(M)$ or $S(P)$: the score of a DAG $G$, a MEC $M$, or a CPDAG $P$. We hide the dependence in $\mathbf{D}$ for simplicity.
- $G + (x \to y)$: the DAG obtained by adding the edge $x \to y$ to $G$. It is undefined if $x \to y$ is already in $G$ or if the resulting graph is not a DAG.
- $G - (x \to y)$: the DAG obtained by removing the edge $x \to y$. It is undefined if $x \to y$ is not in $G$.
- $G \circlearrowleft (x \to y)$: the DAG obtained by reversing the edge $x \to y$. It is undefined if $x \to y$ is not in $G$ or if the resulting graph is not a DAG.
- An edge is *compelled* in a MEC $M$ if it is always directed in the same direction in all DAGs in $M$. By definition, the compelled edges of $M$ are exactly the directed edges of $M$'s canonical PDAG.
- An edge is *reversible* in a MEC $M$ if it is directed in one direction in some DAGs in $M$ and the other direction in other DAGs in $M$. By definition, the reversible edges of $M$ are exactly the undirected edges of $M$'s canonical PDAG.

## B.1 NAVIGATING THE SPACE OF MECS

GES explores the space of MECs by iteratively going from one MEC to the other, each time selecting one with a higher score. GES defines a set of possible candidates that can be reached from a given MEC. This defines the search space of GES [Chickering, 2002b].

For a MEC $M$, define:

- Insertions: $\mathcal{I}(M) = \{M' \in \mathcal{M} \mid \exists G \in M, \exists (x, y) \in [d]^2, G + (x \to y) \in M'\}$.
- Deletions: $\mathcal{D}(M) = \{M' \in \mathcal{M} \mid \exists G \in M, \exists (x, y) \in [d]^2, G - (x \to y) \in M'\}$.
- Reversals : $\mathcal{R}_c(M) = \{M' \in \mathcal{M} \mid \exists G \in M, \exists (x, y) \in [d]^2, (x, y) \text{ is compelled in } P, G \circlearrowleft (x \to y) \in M'\}$.
- Reversals: $\mathcal{R}_r(M) = \{M' \in \mathcal{M} \mid \exists G \in M, \exists (x, y) \in [d]^2, (x, y) \text{ is reversible in } P, G \circlearrowleft (x \to y) \in M'\}$.

The original GES algorithm first navigates through MECs only using $\mathcal{I}$, and then only using $\mathcal{D}$. GIES proposes to add a last step and navigate through MECs using only $\mathcal{R}_c \cup \mathcal{R}_r$ (after $\mathcal{I}$ and $\mathcal{D}$) [Hauser and Bühlmann, 2012].

In contrast, XGES navigates through MECs using simultaneously $\mathcal{I}$, $\mathcal{D}$ and $\mathcal{R}_c$. The XGES heuristics favor using $\mathcal{D}$, then $\mathcal{R}_c$, and finally $\mathcal{I}$ (see Algorithm 2).

**Remark 3.** *XGES could also use $\mathcal{R}_r$, we leave this for future work.*

## B.2 THEORETICAL GUARANTEES OF GES

We reformulate the results in [Chickering, 2002b]. Given a distribution $p^*$ faithful to a graph $G^*$ with MEC $M^*$. GES is *correct* if it returns $M^*$, the MEC of $G^*$. It is the MEC formed by all the DAGs faithful to $p^*$.

**Theorem 6** ([Chickering, 2002b, Lemma 9]). *If no candidate MECs in $\mathcal{I}(M)$ can increase the score $S$, then $M$ has a special structure: all the independencies in $M$ are also independencies in $p^*$. We write it:*

$$\max_{M' \in \mathcal{I}(M)} S(M') \le S(M) \Rightarrow P1(M; p^*),$$

*where $P1(M; p^*)$ is the proposition that all the independencies in $M$ are also independencies in $p^*$.*

Note that $p^*$ might have more independencies than $M$, i.e. $p^*$ is not necessarily faithful to $M$. This is because $M$ might have superfluous edges. For example, the MEC of the complete DAGs satisfies $\texttt{P1}(M; p^*)$.

**Theorem 7** ([Chickering, 2002b, Lemma 10]). *If all the independencies in $M$ are also independencies in $p^*$, then the same is true for all the MECs in $\mathcal{D}(M)$ that have a higher score than $M$. We write it:*

$$\texttt{P1}(M; p^*) \Rightarrow (\forall M' \in \mathcal{D}(M), S(M') \geq S(M) \Rightarrow \texttt{P1}(M'; p^*)).$$

**Theorem 8** ([Chickering, 2002b, Lemma 10]). *If all the independencies in $M$ are also independencies in $p^*$, and if no candidate MECs in $\mathcal{D}(M)$ can increase the score $S$, then $M = M^*$. We write it:*

$$\texttt{P1}(M; p^*) \wedge \left[ \left( \max_{M' \in \mathcal{D}(M)} S(M') \right) \leq S(M) \right] \Rightarrow \texttt{P2}(M; p^*),$$

*where $\texttt{P2}(M; p^*)$ is the proposition that $M = M^*$ or equivalently that $p^*$ is faithful to all the DAGs in $M$*

With Ths. 6 to 8, it follows that GES is correct: it will reach a MEC $M$ at the end of phase 1 such that $\texttt{P1}(M; p^*)$ is true. From there, all MECs visited in phase 2 will also have $\texttt{P1}(M; p^*)$ true, and GES will stop on a MEC $M$ such that $\texttt{P2}(M; p^*)$ is true, i.e. $M = M^*$.

## B.3    THEORETICAL GUARANTEES OF XGES-0 AND XGES

### B.3.1    Theoretical Guarantees of XGES-0

We now prove that XGES-0 is correct.

**Theorem 2.** *For any locally consistent score $S$, the MEC $\hat{M}$ returned by XGES-0 contains the true graph $G^*$.*

*Proof.* We prove that XGES-0 terminates and is correct.

- **Termination.** The algorithm terminates because the search space is finite and the score is non-decreasing at each step.

- **Correctness.** Let $\hat{M}$ be the MEC returned by XGES-0. Since XGES-0 terminated, it means that no candidate MECs in $\mathcal{I}(\hat{M})$ can increase the score $S$. By Th. 6, $\texttt{P1}(\hat{M}; p^*)$ is true. It also means that no candidate MECs in $\mathcal{D}(\hat{M})$ can increase the score $S$. By Th. 8, $\texttt{P2}(\hat{M}; p^*)$ is true. Hence, XGES-0 is correct.

$\square$

### B.3.2 Theoretical Guarantees of XGES

We provide the pseudocode of XGES in Algorithm 3.

---

**Algorithm 3:** XGES

**Data:** Data $\mathbf{D} \in \mathbb{R}^{n \times d}$, score function $S$
**Define:** $\delta_{\mathbf{D},M}(O) = S(\text{Apply}(O, M); \mathbf{D}) - S(M; \mathbf{D})$
**Result:** MEC of $G^*$

1  $M \leftarrow \text{XGES-0}(X, S)$
2  $\mathcal{D} \leftarrow$ all deletes valid for $M$
3  **while** $|\mathcal{D}| > 0$ **do**
4       $D^* \leftarrow \arg\max_{D \in \mathcal{D}}\{\delta_{\mathbf{D},M}(D)\}$
5       $M' \leftarrow \text{Apply}(D^*, M)$
6       $\widetilde{\mathcal{I}} \leftarrow$ all insertions, from any MEC to any MEC, that reinsert the edge deleted by $D^*$
7       $M' \leftarrow \text{XGES-0}^*(X, S, M', \widetilde{\mathcal{I}})$
     /* XGES-0$^*$ is a modified version of XGES from Algorithm 2 that accepts an initial graph $M'$, and a set of forbidden insertions $\widetilde{\mathcal{I}}$      */
8       **if** $S(M'; \mathbf{D}) > S(M; \mathbf{D})$ **then**
9           $M \leftarrow M'$
10          $\mathcal{D} \leftarrow$ all deletes valid for $M$
11      **else**
12          $\mathcal{D} \leftarrow \mathcal{D}\backslash\{D^*\}$

13 **return** $M$

---

**Theorem 3.** *For any score $S$, XGES returns a MEC $\hat{M}$ with a higher or equal score than XGES-0. If $S$ is locally consistent, then $\hat{M}$ contains the true graph $G^*$.*

*Proof.* With a locally consistent score, XGES-0 is correct, so $M$ in line 1 is already the MEC $M^*$ of $G^*$. Such a $M^*$ has the maximum score, so lines 9 and 10 are never executed. Hence, XGES is correct.

For completeness, we provide proof of XGES's termination even when the score is not locally consistent.

**Termination in practice.** Every time XGES replaces $M$ by $M'$, the score of $M$ strictly increases. Since the search space is finite, XGES cannot infinitely replace $M$ by $M'$. But then, the other possibility is to remove $D^*$ from $\mathcal{D}$, which is a finite set. Hence, XGES terminates.

$\square$

## B.4 NAVIGATING THE SPACE OF MECS WITH CPDAGS

We review how to translate the MEC formulation into a CPDAG formulation for the practical implementations of GES and XGES. We recall that each MEC $M$ can be uniquely represented by a CPDAG $P$.

### B.4.1 Original Parametrization

In Section 4.1 we reviewed that given a MEC $M$ represented by the CPDAG $P$, each $M' \in \mathcal{I}(M)$ can be uniquely associated to an operator Insert$(x, y, T)$, where $T$ is a set of nodes, such that applying Insert$(x, y, T)$ to $P$ yields a PDAG $P'$ that represents $M'$ (up to completing it into a CPDAG). The operator Insert$(x, y, T)$ modifies $P$ by adding the edge $x \to y$ and orienting all edges $t - y$ as $t \to y$ for all $t \in T$. Not all Insert$(x, y, T)$ operators can be applied to $P$ and yield a PDAG $P'$ that represents a MEC $M' \in \mathcal{I}(M)$. All the Insert$(x, y, T)$ operators that correspond to a MEC $M' \in \mathcal{I}(M)$ are called valid operators. There is a bijection between $\mathcal{I}(M)$ and the set of valid Insert$(x, y, T)$, and there exist conditions that can be checked on $P$ to determine whether Insert$(x, y, T)$ is valid or not.

The same holds for $\mathcal{D}(M)$, $\mathcal{R}_c(M)$ and $\mathcal{R}_r(M)$. We summarize the operators, their validity conditions, their score and their actions on a CPDAG in Table 2, adapted from Chickering [2002b] for insertion and deletion, and from Hauser and Bühlmann [2012] for reversals.

| Operator | Insert$(x, y, T)$ | Delete$(x, y, H)$ | Reversal$(x, y, T)$ |
|---|---|---|---|
| Conditions | • $x \notin \mathrm{Ad}(y)$ 
 • $T \subset \mathrm{Ne}(y) \setminus \mathrm{Ad}(x)$ 
 • $[\mathrm{Ne}(y) \cap \mathrm{Ad}(x)] \cup T$ is a clique 
 • All semi-directed paths from $y$ to $x$ are blocked by $[\mathrm{Ne}(y) \cap \mathrm{Ad}(x)] \cup T$ | • $x \in \mathrm{Ch}(x) \cup \mathrm{Ne}(x)$ 
 • $H \subset \mathrm{Ne}(y) \cap \mathrm{Ad}(x)$ 
 • $[\mathrm{Ne}(y) \cap \mathrm{Ad}(x)] \setminus H$ is a clique | • $y \in \mathrm{Pa}(x)$ 
 • $T \subset \mathrm{Ne}(y) \setminus \mathrm{Ad}(x)$ 
 • $[\mathrm{Ne}(y) \cap \mathrm{Ad}(x)] \cup T$ is a clique 
 • All semi-directed paths from $y$ to $x$ other than $(y, x)$ are blocked by $[\mathrm{Ne}(y) \cap \mathrm{Ad}(x)] \cup T \cup \mathrm{Ne}(x)$ |
| Score Increase | $s(y, [\mathrm{Ne}(y) \cap \mathrm{Ad}(x)] \cup T \cup \mathrm{Pa}(y) \cup \{x\})$ 
 $-s(y, [\mathrm{Ne}(y) \cap \mathrm{Ad}(x)] \cup T \cup \mathrm{Pa}(y))$ | $s(y, [\mathrm{Ne}(y) \cap \mathrm{Ad}(x)] \setminus H \cup \mathrm{Pa}(y) \setminus \{x\}) -$ 
 $s(y, [\mathrm{Ne}(y) \cap \mathrm{Ad}(x)] \setminus H \cup \mathrm{Pa}(y) \cup \{x\})$ | $s(y, [\mathrm{Ne}(y) \cap \mathrm{Ad}(x)] \cup T \cup \mathrm{Pa}(y) \cup \{x\})$ 
 $-s(y, [\mathrm{Ne}(y) \cap \mathrm{Ad}(x)] \cup T \cup \mathrm{Pa}(y))$ 
 $+s(x, \mathrm{Pa}(x) \setminus \{y\}) - s(x, \mathrm{Pa}(x))$ |
| Actions | • Add $x \to y$ to $P$. 
 • For all $t \in T$, orient $t - y$ as $t \to y$. | • Remove $x \to y$ (or $x - y$) from $P$. 
 • Orient all edges $y - h$ as $h \to y$ for $h \in H$. 
 • Orient all edges $x - h$ as $h \to x$ for $h \in H$. | • Reverse $x \to y$ into $x \leftarrow y$. 
 • For all $t \in T$, orient $t - y$ as $t \to y$. |

Table 2: Summary of the operators and their conditions as described in [Chickering, 2002b]. The conditions for Insert and Delete are described in Chickering [2002b, Definition 12, Definition 13, Theorem 15, Theorem 17, Table 1], and the score increase in Chickering [2002b, Corollary 16, Corollary 18]. The conditions for Reversal are described in Hauser and Bühlmann [2012, Proposition 34], and the score increase in Hauser and Bühlmann [2012, Corollary 36].

### B.4.2 XGES Parametrization

We proposed a slightly different parametrization of the operators with adapted conditions. We recall from Section 4.2.1 that with the original parametrization, the scores of the operators depend on which PDAG they are applied to. With the goal of an efficient implementation that caches the score of valid operators to avoid recomputation, it is important and convenient for each operator to have a unique score, agnostic of the PDAG it is applied to.

We described the new parametrization for the Insert operator in Section 4.2.1. We describe the Delete and Reverse operator in Table 3 hereafter. For the Delete operator, we also replace the set $H$ by the set $C$, its complement in $\mathrm{Ne}(y) \cap \mathrm{Ad}(x)$.

| $\mathrm{Insert}(x, y, T; E)$ | $\mathrm{Delete}(x, y, C; E)$ | $\mathrm{Reverse}(x, y, T, E, F)$ |
|---|---|---|
| | | **R1**: $y \in \mathrm{Pa}(x)$ |
| **I1**: $y \notin \mathrm{Ad}(x)$ | | **R2**: $T \subset \mathrm{Ne}(y) \setminus \mathrm{Ad}(x)$ |
| **I2**: $T \subset \mathrm{Ne}(y) \setminus \mathrm{Ad}(x)$ | **D1**: $y \in \mathrm{Ch}(x) \cup \mathrm{Ne}(x)$ | **R3**: $(\mathrm{Ne}(y) \cap \mathrm{Ad}(x)) \cup T$ is a clique |
| **I3**: $(\mathrm{Ne}(y) \cap \mathrm{Ad}(x)) \cup T$ is a clique | **D2**: $C \subset \mathrm{Ne}(y) \cap \mathrm{Ad}(x)$ | **R4**: All semi-directed paths from $y$ to $x$ |
| **I4**: All semi-directed paths from $y$ to $x$ have a node in $(\mathrm{Ne}(y) \cap \mathrm{Ad}(x)) \cup T$ | **D3**: $C$ is a clique | not using edge $y \to x$ have a node in $(\mathrm{Ne}(y) \cap \mathrm{Ad}(x)) \cup T \cup \mathrm{Ne}(x)$ |
| **I5**: $E = (\mathrm{Ne}(y) \cap \mathrm{Ad}(x)) \cup T \cup \mathrm{Pa}(y)$ | **D4**: $E = C \cup \mathrm{Pa}(y)$ | **R5**: $E = (\mathrm{Ne}(y) \cap \mathrm{Ad}(x)) \cup T \cup \mathrm{Pa}(y)$ |
| | | **R6**: $F = \mathrm{Pa}(x)$ |
| $\delta = s(y, E \cup \{x\}) - s(y, E)$ | $\delta = s(y, E \cup \{x\}) - s(y, E \setminus \{x\})$ | $\delta = s(y, E \cup \{x\}) - s(y, E) + s(x, F \setminus \{y\}) - s(x, F)$ |

Table 3: Parametrization of operators by XGES with their validity conditions and score. This parametrization renders the score of each operator invariant to the CPDAG it is applied to.

We study how each validity condition described in Table 3 can become true after each type of edge update in a PDAG. We only need to consider single-edge updates because of Th. 5.

**Edge Updates**  There are seven types of edge updates: one of $a\ \ b$, $a - b$, or $a \to b$ becomes another one ($6 = 3*2$); and the reversal $a \to b$ into $a \leftarrow b$ (which happens only after applying a reversal operator).

For each of these edge updates, we study how they affect the validity conditions of each operator. We summarize the results in Table 4 and provide the detailed proofs in the following sections. We further aggregate the results from Table 4 into Table 5.

| | $a\ \ b$ → $a - b$ | $a\ \ b$ → $a \to b$ | $a - b$ → $a\ \ b$ | $a - b$ → $a \to b$ | $a \to b$ → $a\ \ b$ | $a \to b$ → $a - b$ | $a \to b$ → $a \leftarrow b$ |
|---|---|---|---|---|---|---|---|
| **I1** | $\varnothing$ | $\varnothing$ | $\{a,b\} = \{x,y\}$ | $\varnothing$ | $\{a,b\} = \{x,y\}$ | $\varnothing$ | $\varnothing$ |
| **I2** | $\begin{cases} \bar a = y \\ \bar b \notin \mathrm{Ad}(x) \end{cases}$ | $\varnothing$ | $\begin{cases} \bar a = x \\ \bar b \in \mathrm{Ne}(y) \end{cases}$ | $\varnothing$ | $\begin{cases} \bar a = x \\ \bar b \in \mathrm{Ne}(y) \end{cases}$ | $\begin{cases} \bar a = y \\ \bar b \notin \mathrm{Ad}(x) \end{cases}$ | $\varnothing$ |
| **I3**[I2] | $\{a,b\} \subset \mathrm{Ne}(y)$ | $\{a,b\} \subset \mathrm{Ne}(y)$ | $\begin{cases} \bar a = y \\ \bar b \in \mathrm{Ad}(x) \end{cases}$ or $\begin{cases} \bar a = x \\ \bar b \in \mathrm{Ne}(y) \end{cases}$ | $\begin{cases} \bar a = y \\ \bar b \in \mathrm{Ad}(x) \end{cases}$ | $\begin{cases} \bar a = x \\ \bar b \in \mathrm{Ne}(y) \end{cases}$ | $\varnothing$ | $\varnothing$ |
| **I4** | $\begin{cases} \bar a = y \\ \bar b \in \mathrm{Ad}(x) \end{cases}$ or $\begin{cases} \bar a = x \\ \bar b \in \mathrm{Ne}(y) \end{cases}$ | $\begin{cases} \bar a = x \\ \bar b \in \mathrm{Ne}(y) \end{cases}$ | $\mathrm{SD}(x,y; \bar a, \bar b)$ | $\mathrm{SD}(x,y; b,a)$ | $\mathrm{SD}(x,y; a,b)$ | $\begin{cases} \bar a = y \\ \bar b \in \mathrm{Ad}(x) \end{cases}$ | $\mathrm{SD}(x,y; a,b)$ |
| **I5** | $\begin{cases} \bar a = y \\ \bar b \in \mathrm{Ad}(x) \end{cases}$ or $\begin{cases} \bar a = x \\ \bar b \in \mathrm{Ne}(y) \end{cases}$ | $b = y$ or $\begin{cases} \bar a = x \\ \bar b \in \mathrm{Ne}(y) \end{cases}$ | $\begin{cases} \bar a = y \\ \bar b \in \mathrm{Ad}(x) \end{cases}$ or $\begin{cases} \bar a = x \\ \bar b \in \mathrm{Ne}(y) \end{cases}$ | $b = y$ or $\begin{cases} \bar a = y \\ \bar b \in \mathrm{Ad}(x) \end{cases}$ | $b = y$ or $\begin{cases} \bar a = x \\ \bar b \in \mathrm{Ne}(y) \end{cases}$ | $b = y$ or $\begin{cases} \bar a = y \\ \bar b \in \mathrm{Ad}(x) \end{cases}$ | $\bar a = y$ |
| **D1** | $\{a,b\} = \{x,y\}$ | $(a,b) = (x,y)$ | $\varnothing$ | $\varnothing$ | $\varnothing$ | $(a,b) = (y,x)$ | $(a,b) = (y,x)$ |
| **D2** | $\begin{cases} \bar a = y \\ \bar b \in \mathrm{Ad}(x) \end{cases}$ or $\begin{cases} \bar a = x \\ \bar b \in \mathrm{Ne}(y) \end{cases}$ | $\begin{cases} \bar a = x \\ \bar b \in \mathrm{Ne}(y) \end{cases}$ | $\varnothing$ | $\varnothing$ | $\varnothing$ | $\begin{cases} \bar a = y \\ \bar b \in \mathrm{Ad}(x) \end{cases}$ | $\varnothing$ |
| **D3**[D2] | $\{a,b\} \subset \mathrm{Ne}^u(y) \cap \mathrm{Ad}^u(x)$ | $\{a,b\} \subset \mathrm{Ne}^u(y) \cap \mathrm{Ad}^u(x)$ | $\varnothing$ | $\varnothing$ | $\varnothing$ | $\varnothing$ | $\varnothing$ |
| **D4** | $\varnothing$ | $b = y$ | $\varnothing$ | $b = y$ | $b = y$ | $b = y$ | $\bar a = y$ |
| **R1** | $\varnothing$ | $(a,b) = (y,x)$ | $\varnothing$ | $(a,b) = (y,x)$ | $\varnothing$ | $\varnothing$ | $(a,b) = (x,y)$ |
| **R2** | See **I2** | See **I2** | See **I2** | See **I2** | See **I2** | See **I2** | See **I2** |
| **R3**[R2] | See **I3** | See **I3** | See **I3** | See **I3** | See **I3** | See **I3** | See **I3** |
| **R4** | See **I4** or $\bar a = x$ | See **I4** | See **I4** | See **I4** | See **I4** | See **I4** or $x = \bar a$ | See **I4** |
| **R5** | See **I5** | See **I5** | See **I5** | See **I5** | See **I5** | See **I5** | See **I5** |
| **R6** | $\varnothing$ | $b = x$ | $\varnothing$ | $b = x$ | $b = x$ | $b = x$ | $\bar a = x$ |

Table 4: For each type of edge update involving an edge $(a,b)$, we list necessary conditions for each validity conditions of operators $\mathrm{Insert}(x,y,T,E)$, $\mathrm{Delete}(x,y,C,E)$, and $\mathrm{Reverse}(x,y,T,E,F)$ to become valid. The notation $\mathrm{Condition}(\bar a, \bar b)$ is a shorthand for $\mathrm{Condition}(a,b) \vee \mathrm{Condition}(b,a)$. The notation $\mathrm{SD}(x,y;a,b)$ is a shorthand for the necessary condition: $(a,b)$, in that order, is on a semi-directed path from $y$ to $x$. All operators $\mathrm{Pa}$, $\mathrm{Ne}$, $\mathrm{Ad}$ and $\mathrm{SD}(x,y;a,b)$ are computed with respect to the PDAG before the edge update. All operators $\mathrm{Pa}^u$, $\mathrm{Ne}^u$, $\mathrm{Ad}^u$ are computed with respect to the PDAG after the edge update.

Table 5 rewrites Table 4 with the statements to be conditions centered around $x$ and $y$, and aggregate all the necessary conditions together. Whenever a single edge $(a,b)$ is updated, only the Insert operators satisfying the condition **I-any** can become valid and need to be checked. The same holds for Delete with **D-any**, and Reverse with **R-any**.

## B.5.1   I1

**Operator Update 1** (Updates on I1)**.** *Assume* $\mathbf{I1}(x,y,T;E)$ *is false and becomes true after an update involving* $(a,b)$. *Then,*

- *the update cannot be* $U1 : (a\ \ b) \rightsquigarrow (a - b)$,

| | $\begin{array}{c}a \quad b\\ a-b\end{array}$ | $\begin{array}{c}a \quad b\\ a \to b\end{array}$ | $\begin{array}{c}a-b\\ a \quad b\end{array}$ | $\begin{array}{c}a-b\\ a \to b\end{array}$ | $\begin{array}{c}a \to b\\ a \quad b\end{array}$ | $\begin{array}{c}a \to b\\ a-b\end{array}$ | $\begin{array}{c}a \to b\\ a \leftarrow b\end{array}$ |
|---|---|---|---|---|---|---|---|
| **I1** | $\varnothing$ | $\varnothing$ | $\{x,y\}=\{a,b\}$ | $\varnothing$ | $\{x,y\}=\{a,b\}$ | $\varnothing$ | $\varnothing$ |
| **I2** | $y=\bar a$ | $\varnothing$ | $\begin{cases}x=\bar a\\ y\in \mathrm{Ne}(\bar b)\end{cases}$ | $\varnothing$ | $\begin{cases}x=\bar a\\ y\in \mathrm{Ne}(\bar b)\end{cases}$ | $y=\bar a$ | $\varnothing$ |
| **I3**[I2] | $y\in \mathrm{Ne}(a)\cap \mathrm{Ne}(b)$ | $y\in \mathrm{Ne}(a)\cap \mathrm{Ne}(b)$ | $\begin{cases}y=\bar a\\ x\in \mathrm{Ad}(\bar b)\end{cases}$ or $\begin{cases}x=\bar a\\ y\in \mathrm{Ne}(\bar b)\end{cases}$ | $\begin{cases}y=\bar a\\ x\in \mathrm{Ad}(\bar b)\end{cases}$ | $\begin{cases}x=\bar a\\ y\in \mathrm{Ne}(\bar b)\end{cases}$ | $\varnothing$ | $\varnothing$ |
| **I4** | $\begin{cases}y=\bar a\\ x\in \mathrm{Ad}(\bar b)\end{cases}$ or $\begin{cases}x=\bar a\\ y\in \mathrm{Ad}(\bar b)\end{cases}$ | $\begin{cases}x=\bar a\\ y\in \mathrm{Ne}(\bar b)\end{cases}$ | $\mathrm{SD}(x,y;\bar a,\bar b)$ | $\mathrm{SD}(x,y;b,a)$ | $\mathrm{SD}(x,y;a,b)$ | $\begin{cases}y=\bar a\\ x\in \mathrm{Ad}(\bar b)\end{cases}$ | $\mathrm{SD}(x,y;a,b)$ |
| **I5** | $\begin{cases}y=\bar a\\ x\in \mathrm{Ad}(\bar b)\end{cases}$ or $\begin{cases}x=\bar a\\ y\in \mathrm{Ne}(\bar b)\end{cases}$ | $y=b$ or $\begin{cases}x=\bar a\\ y\in \mathrm{Ne}(\bar b)\end{cases}$ | $\begin{cases}y=\bar a\\ x\in \mathrm{Ad}(\bar b)\end{cases}$ or $\begin{cases}x=\bar a\\ y\in \mathrm{Ne}(\bar b)\end{cases}$ | $\begin{cases}y=b\text{ or}\\ y=\bar a\\ x\in \mathrm{Ad}(\bar b)\end{cases}$ | $\begin{cases}y=b\text{ or}\\ x=\bar a\\ y\in \mathrm{Ne}(\bar b)\end{cases}$ | $\begin{cases}y=b\text{ or}\\ y=\bar a\\ x\in \mathrm{Ad}(\bar b)\end{cases}$ | $y=\bar a$ |
| **I-any** | • $y\in\{a,b\}$ 
 • $y\in \mathrm{Ne}(a)\cap \mathrm{Ne}(b)$ 
 • $x=a$ and $y\in \mathrm{Ne}(b)$ 
 • $x=b$ and $y\in \mathrm{Ne}(a)$ | • $y=b$ 
 • $y\in \mathrm{Ne}(a)\cap \mathrm{Ne}(b)$ 
 • $x=a$ and $y\in \mathrm{Ne}(b)$ 
 • $x=b$ and $y\in \mathrm{Ne}(a)$ | • $x=a$ and $y\in \mathrm{Ne}(b)\cup\{b\}$ 
 • $x=b$ and $y\in \mathrm{Ne}(a)\cup\{a\}$ 
 • $y=a$ and $x\in \mathrm{Ad}(b)$ 
 • $y=b$ and $x\in \mathrm{Ad}(a)$ 
 • $\mathrm{SD}(x,y;a,b)$ 
 • $\mathrm{SD}(x,y;b,a)$ | • $y=a$ and $x\in \mathrm{Ad}(b)$ 
 • $y=b$ 
 • $\mathrm{SD}(x,y;b,a)$ | • $y=b$ 
 • $x=a$ and $y\in \mathrm{Ne}(b)\cup\{b\}$ 
 • $x=b$ and $y\in \mathrm{Ne}(a)\cup\{a\}$ 
 • $\mathrm{SD}(x,y;a,b)$ | • $y\in\{a,b\}$ | • $y\in\{a,b\}$ 
 • $\mathrm{SD}(x,y;a,b)$ |
| **D1** | $\{x,y\}=\{a,b\}$ | $(x,y)=(a,b)$ | $\varnothing$ | $\varnothing$ | $\varnothing$ | $(x,y)=(b,a)$ | $(x,y)=(b,a)$ |
| **D2** | $\begin{cases}y=\bar a\\ x\in \mathrm{Ad}(\bar b)\end{cases}$ or $\begin{cases}x=\bar a\\ y\in \mathrm{Ne}(\bar b)\end{cases}$ | $\begin{cases}x=\bar a\\ y\in \mathrm{Ne}(\bar b)\end{cases}$ | $\varnothing$ | $\varnothing$ | $\varnothing$ | $\begin{cases}y=\bar a\\ x\in \mathrm{Ad}(\bar b)\end{cases}$ | $\varnothing$ |
| **D3**[D2] | $\begin{cases}x\in \mathrm{Ad}^u(a)\cap \mathrm{Ad}^u(b)\\ y\in \mathrm{Ne}^u(a)\cap \mathrm{Ne}^u(b)\end{cases}$ | $\begin{cases}x\in \mathrm{Ad}^u(a)\cap \mathrm{Ad}^u(b)\\ y\in \mathrm{Ne}^u(a)\cap \mathrm{Ne}^u(b)\end{cases}$ | $\varnothing$ | $\varnothing$ | $\varnothing$ | $\varnothing$ | $\varnothing$ |
| **D4** | $\varnothing$ | $y=b$ | $\varnothing$ | $y=b$ | $y=b$ | $y=b$ | $y\in\{a,b\}$ |
| **D-any** | • $y\in\{a,b\}$ 
 • $x\in\{a,b\}$ 
 • $x\in \mathrm{Ad}(a)\cap \mathrm{Ad}(b)$ and $y\in \mathrm{Ne}(a)\cap \mathrm{Ne}(b)$ | • $y=b$ 
 • $x\in\{a,b\}$ 
 • $x\in \mathrm{Ad}(a)\cap \mathrm{Ad}(b)$ and $y\in \mathrm{Ne}(a)\cap \mathrm{Ne}(b)$ | $\varnothing$ | • $y=b$ | • $y=b$ | • $y\in\{a,b\}$ | • $y\in\{a,b\}$ |
| **R-any** | • $y\in\{a,b\}$ 
 • $y\in \mathrm{Ne}(a)\cap \mathrm{Ne}(b)$ 
 • $x\in\{a,b\}$ | • $y=b$ 
 • $y\in \mathrm{Ne}(a)\cap \mathrm{Ne}(b)$ 
 • $x=a$ and $y\in \mathrm{Ne}(b)$ 
 • $x=b$ | • $x=a$ and $y\in \mathrm{Ne}(b)\cup\{b\}$ 
 • $x=b$ and $y\in \mathrm{Ne}(a)\cup\{a\}$ 
 • $y=a$ and $x\in \mathrm{Ad}(b)$ 
 • $y=b$ and $x\in \mathrm{Ad}(a)$ 
 • $\mathrm{SD}(x,y;a,b)$ 
 • $\mathrm{SD}(x,y;b,a)$ | • $y=a$ and $x\in \mathrm{Ad}(b)$ 
 • $y=b$ 
 • $\mathrm{SD}(x,y;b,a)$ 
 • $x=b$ | • $y=b$ 
 • $x=a$ and $y\in \mathrm{Ne}(b)\cup\{b\}$ 
 • $x=b$ 
 • $\mathrm{SD}(x,y;a,b)$ | • $y\in\{a,b\}$ 
 • $x=b$ | • $y\in\{a,b\}$ 
 • $x\in\{a,b\}$ 
 • $\mathrm{SD}(x,y;a,b)$ |

Table 5: For each type of edge update involving an edge $(a,b)$, we list necessary conditions for each validity conditions of operators Insert$(x,y,T,E)$, Delete$(x,y,C,E)$, and Reverse$(x,y,T,E,F)$ to become valid. The notation $\mathrm{SD}(x,y;a,b)$ is a shorthand for the necessary condition: $(a,b)$, in that order, is on a semi-directed path from $y$ to $x$. All operators Pa, Ne, Ad and $\mathrm{SD}(x,y;a,b)$ are computed with respect to the PDAG before the edge update. The rows **I**-any, **D**-any, and **R**-any aggregate the necessary conditions for each validity condition and express them in a disjunctive form: at least one of the conditions must be true for the operator to become valid.

- *the update cannot be $U2 : (a \quad b) \rightsquigarrow (a \to b)$,*
- *if the update is $U3 : (a-b) \rightsquigarrow (a \quad b)$ then $\{a,b\}=\{x,y\}$,*
- *the update cannot be $U4 : (a-b) \rightsquigarrow (a \to b)$,*
- *if the update is $U5 : (a \to b) \rightsquigarrow (a \quad b)$ then $\{a,b\}=\{x,y\}$,*
- *the update cannot be $U6 : (a \to b) \rightsquigarrow (a-b)$,*
- *the update cannot be $U7 : (a \to b) \rightsquigarrow (a \leftarrow b)$.*

*Proof.* We recall that $\mathbf{I1}(x,y,T;E)$ is $y\notin \mathrm{Ad}(x)$. So the assumptions are $y\in \mathrm{Ad}(x)$ and $y\notin \mathrm{Ad}^u(x)$. But U1, U2, U4, U6, and U7 do not remove any elements from any $\mathrm{Ad}(x')$ set. So none of them can render $\mathbf{I1}(x,y,T;E)$ true.

U3 and U5 can only remove elements from $\mathrm{Ad}(a)$ or $\mathrm{Ad}(b)$, and do so only by removing $b$ or $a$, respectively. So $\{x,y\}=\{a,b\}$. $\qquad\square$

### B.5.2 I2

We start with a general lemma for I2.

**Lemma 1** (I2 to become true). *Assume $\mathbf{I2}(x, y, T; E)$ is false and becomes true after an edge update. Then (i) $\mathrm{Ne}^u(y)$ gained an element, or (ii) $\mathrm{Ad}^u(x)$ lost an element that was in $\mathrm{Ne}(y)$.*

*Proof.* We recall that $\mathbf{I2}(x, y, T; E)$ is $T \subset \mathrm{Ne}(y) \setminus \mathrm{Ad}(x)$. If $\mathbf{I2}(x, y, T; E)$ changes from false to true then there exists $t \in T$ that was not in $\mathrm{Ne}(y) \setminus \mathrm{Ad}(x)$ and is now in $\mathrm{Ne}^u(y) \setminus \mathrm{Ad}^u(x)$, which writes

$$(t \notin \mathrm{Ne}(y) \vee t \in \mathrm{Ad}(x)) \wedge t \in \mathrm{Ne}^u(y) \wedge t \notin \mathrm{Ad}^u(x).$$

- If $t \in \mathrm{Ne}(y)$ then we must have $t \in \mathrm{Ad}(x)$ and $t \notin \mathrm{Ad}^u(x)$. So $\mathrm{Ad}^u(x)$ lost $t$, which was in $\mathrm{Ne}(y) \cap \mathrm{Ad}(x)$.
- If $t \notin \mathrm{Ne}(y)$, then $\mathrm{Ne}^u(y)$ gained $t$.

In conclusion, either $\mathrm{Ne}^u(y)$ gained an element, or $\mathrm{Ad}^u(x)$ lost an element that was is in $\mathrm{Ne}(y)$.

$\square$

**Operator Update 2** (Updates on I2). *Assume $\mathbf{I2}(x, y, T; E)$ is false and becomes true after an update involving $(a, b)$. Then,*

- *if the update is $U1 : (a \quad b) \rightsquigarrow (a - b)$ then $y \in \{a, b\}$,*
- *the update cannot be $U2 : (a \quad b) \rightsquigarrow (a \to b)$,*
- *if the update is $U3 : (a - b) \rightsquigarrow (a \quad b)$ then $\begin{cases} a = x \\ b \in \mathrm{Ne}(y) \end{cases}$ or $\begin{cases} b = x \\ a \in \mathrm{Ne}(y) \end{cases}$,*
- *the update cannot be $U4 : (a - b) \rightsquigarrow (a \to b)$,*
- *if the update is $U5 : (a \to b) \rightsquigarrow (a \quad b)$ then $\begin{cases} a = x \\ b \in \mathrm{Ne}(y) \end{cases}$ or $\begin{cases} b = x \\ a \in \mathrm{Ne}(y) \end{cases}$,*
- *if the update is $U6 : (a \to b) \rightsquigarrow (a - b)$ then $y \in \{a, b\}$,*
- *the update cannot be $U7 : (a \to b) \rightsquigarrow (a \leftarrow b)$.*

*Proof.* According to Lemma 1, either $\mathrm{Ne}^u(y)$ gained an element or $\mathrm{Ad}^u(x)$ lost an element (that was in $\mathrm{Ne}(y)$).

We now study the necessary conditions for each update, if it was applied and made $\mathbf{I2}(x, y, T; E)$ become true.

- The updates U1 and U6 can only add elements to $\mathrm{Ne}^u(a)$ or $\mathrm{Ne}^u(b)$ and not remove any element to any $\mathrm{Ad}^u(x')$. So $\mathrm{Ne}^u(y)$ gained an element and $y \in \{a, b\}$.
- The updates U2, U4, and U7 do not add any elements to any $\mathrm{Ne}^u(y')$, and do not remove any elements to any $\mathrm{Ad}^u(x')$. So none of them can make $\mathbf{I2}(x, y, T; E)$ become true.
- The updates U3 and U5 can only remove elements from $\mathrm{Ad}^u(a)$ or $\mathrm{Ad}^u(b)$ and not add any element to any $\mathrm{Ne}^u(y')$. So $\mathrm{Ad}^u(x)$ lost an element and $x \in \{a, b\}$. If $x = a$ (resp. $x = b$) then the lost element must be $b$ (resp. $a$) and so $b \in \mathrm{Ne}(y)$ (resp. $a \in \mathrm{Ne}(y)$).

$\square$

### B.5.3 I3

We start with a general lemma for I3.

**Lemma 2** (I3 to become true). *Assume $\mathbf{I3}(x, y, T; E)$ is false and becomes true after an edge update about $(a, b)$. Further, assume that $\mathbf{I2}(x, y, T; E)$ is true after the update (regardless of its status before the update). Then either (i) $\{a, b\} \subset \mathrm{Ne}(y)$ and the update rendered $a, b$ adjacent, or (ii) $\mathrm{Ne}^u(y)$ lost an element that was in $\mathrm{Ne}(y) \cap \mathrm{Ad}(x)$, or (iii) $\mathrm{Ad}^u(x)$ lost an element that was in $\mathrm{Ne}(y) \cap \mathrm{Ad}(x)$.*

*Proof.* We recall that **I3**$(x, y, T; E)$ is $[\mathrm{Ne}(y) \cap \mathrm{Ad}(x)] \cup T$ is a clique, and that **I2**$(x, y, T; E)$ is $T \subset \mathrm{Ne}(y) \backslash \mathrm{Ad}(x)$. So the assumptions are $[\mathrm{Ne}(y) \cap \mathrm{Ad}(x)] \cup T$ is not a clique (in the pre-update PDAG) meanwhile $[\mathrm{Ne}^u(y) \cap \mathrm{Ad}^u(x)] \cup T$ is a clique (in the post-update PDAG), and $T \subset \mathrm{Ne}^u(y) \backslash \mathrm{Ad}^u(x)$.

Since $[\mathrm{Ne}(y) \cap \mathrm{Ad}(x)] \cup T$ is not a clique, it must contain two nodes $c, d$ that are not connected in the pre-update PDAG.

We distinguish two cases:

- If $\{c, d\} \subset [\mathrm{Ne}^u(y) \cap \mathrm{Ad}^u(x)] \cup T$, then the update must have connected $c$ and $d$. So $c$ and $d$ are $a$ and $b$. Also since $T \subset \mathrm{Ne}^u(y) \backslash \mathrm{Ad}^u(x)$, then $\{a, b\} \subset [\mathrm{Ne}^u(y) \cap \mathrm{Ad}^u(x)] \cup T \subset \mathrm{Ne}^u(y)$. Finally, an update can only change one edge at a time, so $\mathrm{Ne}^u(y) = \mathrm{Ne}(y)$ (since $a$ and $b$ are not $y$ as $y$ cannot be a neighbor of itself). Hence, $\{a, b\} \subset \mathrm{Ne}(y)$ and $a$ and $b$ became adjacent.

- Else, $c$ or $d$ has been removed from $[\mathrm{Ne}^u(y) \cap \mathrm{Ad}^u(x)] \cup T$ during the update. Without loss of generality, assume $c$ was removed. Since $T$ does not change, then $c$ was removed from $[\mathrm{Ne}(y) \cap \mathrm{Ad}(x)]$. So $\mathrm{Ne}^u(y)$ or $\mathrm{Ad}^u(x)$ lost an element that was in $\mathrm{Ne}(y) \cap \mathrm{Ad}(x)$.

Hence, $[\{a, b\} \subset \mathrm{Ne}(y)$ and $a$ and $b$ became adjacent$]$, or $\mathrm{Ne}(y)$ lost an element that was in $\mathrm{Ne}(y) \cap \mathrm{Ad}(x)$ or $\mathrm{Ad}(x)$ lost an element that was in $\mathrm{Ne}(y) \cap \mathrm{Ad}(x)$. $\qquad\square$

**Operator Update 3** (Updates on I3). *Assume* **I3**$(x, y, T; E)$ *is false and becomes true after an update involving* $(a, b)$. *Further, assume that* **I2**$(x, y, T; E)$ *is true after the update. Then,*

- *if the update is* $U1 : (a \quad b) \rightsquigarrow (a - b)$ *then* $\{a, b\} \subset \mathrm{Ne}(y)$,
- *if the update is* $U2 : (a \quad b) \rightsquigarrow (a \to b)$ *then* $\{a, b\} \subset \mathrm{Ne}(y)$,
- *if the update is* $U3 : (a - b) \rightsquigarrow (a \quad b)$ *then* $\left\{ \begin{array}{c} a \in \{x, y\} \\ b \in \mathrm{Ne}(y) \cap \mathrm{Ad}(x) \end{array} \right.$ *or* $\left\{ \begin{array}{c} b \in \{x, y\} \\ a \in \mathrm{Ne}(y) \cap \mathrm{Ad}(x) \end{array} \right.$,
- *if the update is* $U4 : (a - b) \rightsquigarrow (a \to b)$ *then* $\left\{ \begin{array}{c} a = y \\ b \in \mathrm{Ne}(y) \cap \mathrm{Ad}(x) \end{array} \right.$ *or* $\left\{ \begin{array}{c} b = y \\ a \in \mathrm{Ne}(y) \cap \mathrm{Ad}(x) \end{array} \right.$,
- *if the update is* $U5 : (a \to b) \rightsquigarrow (a \quad b)$ *then* $\left\{ \begin{array}{c} a = x \\ b \in \mathrm{Ne}(y) \cap \mathrm{Ad}(x) \end{array} \right.$ *or* $\left\{ \begin{array}{c} b = x \\ a \in \mathrm{Ne}(y) \cap \mathrm{Ad}(x) \end{array} \right.$,
- *if the update is* $U6 : (a \to b) \rightsquigarrow (a - b)$ *then it is impossible,*
- *if the update is* $U7 : (a \to b) \rightsquigarrow (a \leftarrow b)$ *then it is impossible.*

*Proof.* According to Lemma 2, either $\{a, b\} \subset \mathrm{Ne}(y)$ and the update rendered $a, b$ adjacent, or $\mathrm{Ne}^u(y)$ lost an element that was in $\mathrm{Ne}(y) \cap \mathrm{Ad}(x)$, or $\mathrm{Ad}^u(x)$ lost an element that was in $\mathrm{Ne}(y) \cap \mathrm{Ad}(x)$.

We now study the necessary conditions for each update, if it was applied and made **I3**$(x, y, T; E)$ become true.

- The updates U1 and U2 can only add elements to sets like $\mathrm{Ne}^u(y)$ or $\mathrm{Ad}^u(x)$, so the only possibility is that $\{a, b\} \subset \mathrm{Ne}(y)$.

- The update U3 does not render any edge adjacent. So by Lemma 2, $\mathrm{Ne}(y)$ or $\mathrm{Ad}(x)$ lost an element $c$ that was in $\mathrm{Ne}(y) \cap \mathrm{Ad}(x)$.

  – If it is $\mathrm{Ne}(y)$ that lost $c$, then we have $\{a, b\} = \{y, c\}$. Without loss of generality, $a = y$ and $b = c$, so $b \in \mathrm{Ne}(y) \cap \mathrm{Ad}(x)$.

  – If it is $\mathrm{Ad}(x)$ that lost $c$, then we have $\{a, b\} = \{x, c\}$. Without loss of generality, $a = x$ and $b = c$, so $b \in \mathrm{Ne}(y) \cap \mathrm{Ad}(x)$.

  So gathering all cases and with generality:

$$\left\{ \begin{array}{c} a \in \{x, y\} \\ b \in \mathrm{Ne}(y) \cap \mathrm{Ad}(x) \end{array} \right. \quad \text{or} \quad \left\{ \begin{array}{c} b \in \{x, y\} \\ a \in \mathrm{Ne}(y) \cap \mathrm{Ad}(x) \end{array} \right.$$

- The update U4 does not render any edge adjacent, does not remove any element from any $\mathrm{Ad}(x')$, but removes elements from $\mathrm{Ne}^u(a)$ or $\mathrm{Ne}^u(b)$ (resp.$b$ or $a$). So by Lemma 2, $a = y$ (resp. $b = y$) and $b \in \mathrm{Ne}(y) \cap \mathrm{Ad}(x)$ (resp. $a \in \mathrm{Ne}(y) \cap \mathrm{Ad}(x)$).

- The update U5 does not render any edge adjacent, does not remove any element from any $\mathrm{Ne}(y')$, but removes elements from $\mathrm{Ad}^u(a)$ or $\mathrm{Ad}^u(b)$ (resp. $b$ or $a$). So by Lemma 2, $a = x$ (resp. $b = x$) and $b \in \mathrm{Ne}(y) \cap \mathrm{Ad}(x)$ (resp. $a \in \mathrm{Ne}(y) \cap \mathrm{Ad}(x)$).

- The updates U6 and U7 cannot remove any element from any $\mathrm{Ne}(y')$ or $\mathrm{Ad}(x')$, and do not make $a$ and $b$ adjacent (they were already adjacent). So by Lemma 2, they cannot make $\mathbf{I3}(x, y, T; E)$ become true.

$\square$

### B.5.4 I4

We start with a general lemma for I4.

**Lemma 3** (I4 to become true). *Assume $\mathbf{I4}(x, y, T; E)$ is false and becomes true after an edge update about $(a, b)$. If the update does not reverse a directed edge, does not direct an undirected edge, and does not delete an edge, then the update must have added an element to $[\mathrm{Ne}(y) \cap \mathrm{Ad}(x)]$.*

*Otherwise, the update invalidated an edge on a semi-directed path from $y$ to $x$ (where invalidated means that the edge $(a, b)$ cannot be traversed from $a$ to $b$ anymore with the semi-directed rules: either $a$ and $b$ are not adjacent anymore, or the edge is now $a \leftarrow b$.).*

*Proof.* We recall that $\mathbf{I4}(x, y, T; E)$ is: all semi-directed paths from $y$ to $x$ have a node in $[\mathrm{Ne}(y) \cap \mathrm{Ad}(x)] \cup T$. If the condition does not hold before the update, then there exists a semi-directed path from $y$ to $x$ with no node in $[\mathrm{Ne}(y) \cap \mathrm{Ad}(x)] \cup T$. We distinguish two cases:

If the update does not remove or reverse any edge, then the semi-directed path is still there after the update. For the condition to become true, the update must have added an element to $[\mathrm{Ne}(y) \cap \mathrm{Ad}(x)] \cup T$ (one element that is on the semi-directed path).

Since $T$ does not change, the update must have added an element to $[\mathrm{Ne}(y) \cap \mathrm{Ad}(x)]$.

Otherwise, the semi-directed path from $y$ to $x$ is not a semi-directed path anymore. So the update invalidated an edge on it: either $a$ and $b$ are not adjacent anymore, or the edge is now $a \leftarrow b$. $\square$

**Operator Update 4** (Updates on I4). *Assume $\mathbf{I4}(x, y, T; E)$ is false and becomes true after an update involving $(a, b)$. Then,*

- *if the update is $U1 : (a \quad b) \rightsquigarrow (a - b)$ then $\begin{cases} b = y \\ a \in \mathrm{Ad}(x) \end{cases}$ or $\begin{cases} b = x \\ a \in \mathrm{Ne}(y) \end{cases}$ or $\begin{cases} a = y \\ b \in \mathrm{Ad}(x) \end{cases}$ or $\begin{cases} a = x \\ b \in \mathrm{Ne}(y) \end{cases}$,*

- *if the update is $U2 : (a \quad b) \rightsquigarrow (a \rightarrow b)$ then $\begin{cases} b = x \\ a \in \mathrm{Ne}(y) \end{cases}$ or $\begin{cases} a = x \\ b \in \mathrm{Ne}(y) \end{cases}$.*

- *if the update is $U3 : (a - b) \rightsquigarrow (a \quad b)$ then either $(a, b)$ or $(b, a)$ was on a semi-directed path from $y$ to $x$.*

- *if the update is $U4 : (a - b) \rightsquigarrow (a \rightarrow b)$ then $(b, a)$ was on a semi-directed path from $y$ to $x$.*

- *if the update is $U5 : (a \rightarrow b) \rightsquigarrow (a \quad b)$ then $(a, b)$ was on a semi-directed path from $y$ to $x$.*

- *if the update is $U6 : (a \rightarrow b) \rightsquigarrow (a - b)$ then $\begin{cases} b = y \\ a \in \mathrm{Ad}(x) \end{cases}$ or $\begin{cases} a = y \\ b \in \mathrm{Ad}(x) \end{cases}$.*

- *if the update is $U7 : (a \rightarrow b) \rightsquigarrow (a \leftarrow b)$ then $(a, b)$ was on a semi-directed path from $y$ to $x$.*

*Proof.* Assume $\mathbf{I4}(x, y, T; E)$ is false and becomes true after an update involving $(a, b)$. U1, U2, and U6 do not reverse any directed edge, do not direct any undirected edge, and do not delete any edge. So by Lemma 3, these updates must have added an element to $[\mathrm{Ne}(y) \cap \mathrm{Ad}(x)]$. Without loss of generality for now, assume $a$ was added. Notice that $a$ cannot have been added to both $\mathrm{Ne}(y)$ and $\mathrm{Ad}(x)$ (otherwise $y = b$ and $x = b$, yet $x \neq y$), so $a$ was already in one of them before the update.

- If the update is U1 then $a$ can have been added to $\mathrm{Ne}(y)$ or $\mathrm{Ad}(x)$, and already present in the other one. Hence we have, in full generality, $\begin{cases} b = y \\ a \in \mathrm{Ad}(x) \end{cases}$ or $\begin{cases} b = x \\ a \in \mathrm{Ne}(y) \end{cases}$ or $\begin{cases} a = y \\ b \in \mathrm{Ad}(x) \end{cases}$ or $\begin{cases} a = x \\ b \in \mathrm{Ne}(y) \end{cases}$.

- If the update is U2 then $a$ can only have been added to $\mathrm{Ad}(x)$, and so already present in $\mathrm{Ne}(y)$. Hence we have, in full generality, $\begin{cases} b = x \\ a \in \mathrm{Ne}(y) \end{cases}$ or $\begin{cases} a = x \\ b \in \mathrm{Ne}(y) \end{cases}$.

- If the update is U6 then $a$ can only have been added to $\mathrm{Ne}(y)$, and so already present in $\mathrm{Ad}(x)$. Hence we have, in full generality, $\begin{cases} b = y \\ a \in \mathrm{Ad}(x) \end{cases}$ or $\begin{cases} a = y \\ b \in \mathrm{Ad}(x) \end{cases}$.

U3, U4, U5, and U7 cannot add any element to $[\mathrm{Ne}(y) \cap \mathrm{Ad}(x)]$. So by Lemma 3, these updates must have invalidated an edge on a semi-directed path from $y$ to $x$.

- If the update is U3 then either $(a, b)$ or $(b, a)$ was on a semi-directed path from $y$ to $x$.
- If the update is U4 then $(b, a)$ was on a semi-directed path from $y$ to $x$.
- If the update is U5 or U7, then $(a, b)$ was on a semi-directed path from $y$ to $x$.

$\square$

So far, all the necessary conditions for the updates were efficient to test, e.g. finding all the insert with $y \in \{a, b\}$.

With **I4** however, we have the condition that $(a, b)$ or $(b, a)$ was on a semi-directed path from $y$ to $x$. This can be inefficient to test. A speed-up can be obtained by proceeding as follows

- Instead of ensuring that $\mathcal{C}$ always contains all valid Insert operators (which means all operators that with **I1**, **I2**, **I3**, **I4**, and **I5** are true), we can ensure that $\mathcal{C}$ always contains all Insert operator for which **I1**, **I2**, **I3**, and **I5** are true.

- Recall that at each step, XGES involves 4 substeps described in Section 4.2. Substep 3 verifies that the operator is valid. If we notice an operator that is invalid because of **I4**, then we can put it aside. Additionally, we save the path from $y$ to $x$ that was rendering the operator invalid.

- The new necessary condition for the operator to be valid is that an edge on the saved path gets removed (or blocked). Whenever we remove or block an edge from the path, we can re-verify the operator.

### B.5.5 I5

We start with a general lemma for I5.

**Lemma 4** (I5 to become true). *Assume* **I5**$(x, y, T; E)$ *is false and becomes true after an edge update about* $(a, b)$*. Then (i)* $\mathrm{Pa}(y)$ *changed, or (ii)* $[\mathrm{Ne}(y) \cap \mathrm{Ad}(x)]$ *changed.*

*Condition (ii) can be further broken down into: (ii.a)* $\mathrm{Ne}^u(y)$ *lost an element that is in* $\mathrm{Ad}(x)$*, or (ii.b)* $\mathrm{Ne}^u(y)$ *gained an element that is in* $\mathrm{Ad}(x)$*, or (ii.c)* $\mathrm{Ne}^u(y)$ *gained an element that is in* $\mathrm{Ad}(x)$*, or (ii.d)* $\mathrm{Ad}^u(x)$ *gained an element that is in* $\mathrm{Ne}(y)$*.*

*Proof.* Assume **I5**$(x, y, T; E)$ is false and becomes true after an edge update about $(a, b)$. Since $E$ and $T$ do not change, we must have $[\mathrm{Ne}^u(y) \cap \mathrm{Ad}^u(x)] \cup \mathrm{Pa}^u(y) \neq [\mathrm{Ne}(y) \cap \mathrm{Ad}(x)] \cup \mathrm{Pa}(y)$. Either $\mathrm{Pa}(y)$ changed or $[\mathrm{Ne}(y) \cap \mathrm{Ad}(x)]$ changed.

Assume $[\mathrm{Ne}(y) \cap \mathrm{Ad}(x)]$ changed. If it lost an element $c \in \mathrm{Ne}(y) \cap \mathrm{Ad}(x)$, then $c$ was removed from $\mathrm{Ne}^u(y)$ or $\mathrm{Ad}^u(x)$. If it gained an element $c \notin \mathrm{Ne}(y) \cap \mathrm{Ad}(x)$, then $c$ was added to $\mathrm{Ne}^u(y)$ or $\mathrm{Ad}^u(x)$. Since $c \in \mathrm{Ne}^u(y) \cap \mathrm{Ad}^u(x)$ and only one of $\mathrm{Ne}^u(y)$ and $\mathrm{Ad}^u(x)$ can change at a time (see proof Operator Update 4), then either $c$ was added to $\mathrm{Ne}^u(y)$ and $c \in \mathrm{Ad}(x)$, or $c$ was added to $\mathrm{Ad}^u(x)$ and $c \in \mathrm{Ne}(y)$. $\square$

**Operator Update 5** (Updates on I5). *Assume* **I5**$(x, y, T; E)$ *is false and becomes true after an update involving* $(a, b)$*. Then,*

- *if the update is* $U1 : (a \quad b) \rightsquigarrow (a - b)$ *then* $\begin{cases} b = y \\ a \in \mathrm{Ad}(x) \end{cases}$ *or* $\begin{cases} b = x \\ a \in \mathrm{Ne}(y) \end{cases}$ *or* $\begin{cases} a = y \\ b \in \mathrm{Ad}(x) \end{cases}$ *or* $\begin{cases} a = x \\ b \in \mathrm{Ad}(y) \end{cases}$,

- *if the update is* $U2 : (a \quad b) \rightsquigarrow (a \rightarrow b)$ *then* $y = b$ *or* $\begin{cases} b = x \\ a \in \mathrm{Ne}(y) \end{cases}$ *or* $\begin{cases} a = x \\ b \in \mathrm{Ne}(y) \end{cases}$,

- *if the update is $U3: (a - b) \rightsquigarrow (a \quad b)$ then* $\begin{cases} b = y \\ a \in \mathrm{Ad}(x) \end{cases}$ *or* $\begin{cases} b = x \\ a \in \mathrm{Ne}(y) \end{cases}$ *or* $\begin{cases} a = y \\ b \in \mathrm{Ad}(x) \end{cases}$ *or* $\begin{cases} a = x \\ b \in \mathrm{Ad}(y) \end{cases}$,

- *if the update is $U4: (a - b) \rightsquigarrow (a \rightarrow b)$ then $y = b$ or* $\begin{cases} a = y \\ b \in \mathrm{Ad}(x) \end{cases}$ *or* $\begin{cases} b = y \\ a \in \mathrm{Ad}(x) \end{cases}$.

- *if the update is $U5: (a \rightarrow b) \rightsquigarrow (a \quad b)$ then $y = b$ or* $\begin{cases} a = x \\ b \in \mathrm{Ne}(y) \end{cases}$ *or* $\begin{cases} b = x \\ a \in \mathrm{Ne}(y) \end{cases}$.

- *if the update is $U6: (a \rightarrow b) \rightsquigarrow (a - b)$ then $y = b$ or* $\begin{cases} a = y \\ b \in \mathrm{Ad}(x) \end{cases}$ *or* $\begin{cases} b = y \\ a \in \mathrm{Ad}(x) \end{cases}$.

- *if the update is $U7: (a \rightarrow b) \rightsquigarrow (a \leftarrow b)$ then $y \in \{a, b\}$.*

*Proof.* According to Lemma 4, either $\mathrm{Pa}(y)$ changed, or $[\mathrm{Ne}(y) \cap \mathrm{Ad}(x)]$ changed.

We now study the necessary conditions for each update, if it was applied and made $\mathbf{I5}(x, y, T; E)$ become true.

- U1 does not change $\mathrm{Pa}(y)$, and cannot remove any element from any $\mathrm{Ne}(y')$ or $\mathrm{Ad}(x')$. So by Lemma 4, (ii.c) or (ii.d) happened. Hence: $\begin{cases} b = y \\ a \in \mathrm{Ad}(x) \end{cases}$ or $\begin{cases} b = x \\ a \in \mathrm{Ne}(y) \end{cases}$ or $\begin{cases} a = y \\ b \in \mathrm{Ad}(x) \end{cases}$ or $\begin{cases} a = x \\ b \in \mathrm{Ne}(y) \end{cases}$.

- U2 changes $\mathrm{Pa}(y)$ if $y = b$. It can also add an element to $\mathrm{Ad}^u(x)$ so: $y = b$ or $\begin{cases} b = x \\ a \in \mathrm{Ne}(y) \end{cases}$ or $\begin{cases} a = x \\ b \in \mathrm{Ne}(y) \end{cases}$.

- U3 does not change $\mathrm{Pa}(y)$, and does not add any element to any $\mathrm{Ne}(y')$ or $\mathrm{Ad}(x')$. So by Lemma 4, (ii.a) or (ii.b) happened. Hence: $\begin{cases} a = x \\ b \in \mathrm{Ne}(y) \end{cases}$ or $\begin{cases} a = y \\ b \in \mathrm{Ad}(x) \end{cases}$ or $\begin{cases} b = x \\ a \in \mathrm{Ne}(y) \end{cases}$ or $\begin{cases} b = y \\ a \in \mathrm{Ad}(x) \end{cases}$.

- U4 changes $\mathrm{Pa}(y)$ if $y = b$. It also removes an element from $\mathrm{Ne}^u(a)$ and $\mathrm{Ne}^u(b)$. So: $y = b$ or $\begin{cases} a = y \\ b \in \mathrm{Ad}(x) \end{cases}$ or $\begin{cases} b = y \\ a \in \mathrm{Ad}(x) \end{cases}$.

- U5 changes $\mathrm{Pa}(y)$ if $y = b$. It also removes an element from $\mathrm{Ad}^u(a)$ and $\mathrm{Ad}^u(b)$. So: $y = b$ or $\begin{cases} a = x \\ b \in \mathrm{Ne}(y) \end{cases}$ or $\begin{cases} b = x \\ a \in \mathrm{Ne}(y) \end{cases}$.

- U6 changes $\mathrm{Pa}(y)$ if $y = b$. It also adds an element to $\mathrm{Ne}^u(a)$ and $\mathrm{Ne}^u(b)$. So: $y = b$ or $\begin{cases} a = y \\ b \in \mathrm{Ad}(x) \end{cases}$ or $\begin{cases} b = y \\ a \in \mathrm{Ad}(x) \end{cases}$.

- U7 changes $\mathrm{Pa}(y)$ if $y \in \{a, b\}$. It does not change any $\mathrm{Ne}(y')$ or $\mathrm{Ad}(x')$. So $y \in \{a, b\}$.

$\square$

### B.5.6  D1

**Operator Update 6** (Updates on D1). *Assume $\mathbf{D1}(x, y, T; E)$ is false and becomes true after an update involving $(a, b)$. Then,*

- *if the update is $U1: (a \quad b) \rightsquigarrow (a - b)$ then $\{a, b\} = \{x, y\}$,*
- *if the update is $U2: (a \quad b) \rightsquigarrow (a \rightarrow b)$ then $(a, b) = (x, y)$,*
- *the update cannot be $U3: (a - b) \rightsquigarrow (a \quad b)$,*
- *the update cannot be $U4: (a - b) \rightsquigarrow (a \rightarrow b)$,*
- *the update cannot be $U5: (a \rightarrow b) \rightsquigarrow (a \quad b)$,*
- *if the update is $U6: (a \rightarrow b) \rightsquigarrow (a - b)$ then $(a, b) = (y, x)$,*
- *if the update is $U7: (a \rightarrow b) \rightsquigarrow (a \leftarrow b)$ then $(a, b) = (y, x)$.*

*Proof.* We recall that $\mathbf{D1}(x, y, C; E)$ is: $y \in \mathrm{Ch}(x) \cup \mathrm{Ne}(x)$. Assume $\mathbf{D1}(x, y, C; E)$ is false and becomes true after an update involving $(a, b)$.

- U3 and U5 cannot add any element to $y$'s children or neighbors. So they cannot make $\mathbf{D1}(x, y, C; E)$ become true.

- U4 does not add any elements to any $\mathrm{Ch}^u(x') \cup \mathrm{Ne}^u(x')$, so it cannot make $\mathbf{D1}(x, y, C; E)$ become true.

- U1 only adds $a$ to $\mathrm{Ne}^u(b) \cup \mathrm{Ch}^u(b)$ and $b$ to $\mathrm{Ne}^u(a) \cup \mathrm{Ch}^u(a)$. So $\{a, b\} = \{x, y\}$.

- U2 only adds $b$ to $\mathrm{Ch}^u(a) \cup \mathrm{Ne}^u(a)$ so $(a, b) = (x, y)$.

- U6 only adds $a$ to $\mathrm{Ch}^u(b) \cup \mathrm{Ne}^u(b)$ so $(a, b) = (y, x)$.

- U7 only adds $a$ to $\mathrm{Ch}^u(b) \cup \mathrm{Ne}^u(b)$ so $(a, b) = (y, x)$.

$\square$

### B.5.7   D2

We start with a general lemma for D2.

**Lemma 5** (D2 to become true). *Assume $\mathbf{D2}(x, y, C; E)$ is false and becomes true after an edge update about $(a, b)$. Then $\mathrm{Ne}^u(y)$ gained an element that was in $\mathrm{Ad}(x)$, or $\mathrm{Ad}^u(x)$ gained an element that was in $\mathrm{Ne}(y)$.*

*Proof.* We recall that $\mathbf{D2}(x, y, C; E)$ is: $C \subset \mathrm{Ne}(y) \cap \mathrm{Ad}(x)$. Assume $\mathbf{D2}(x, y, C; E)$ is false and becomes true after an edge update about $(a, b)$. Since $C$ does not change, we there exists $c \in C$ such that $c \notin \mathrm{Ne}(y) \cap \mathrm{Ad}(x)$ and $c \in \mathrm{Ne}^u(y) \cap \mathrm{Ad}^u(x)$. Then we conclude with a similar reasoning as in Lemma 4. $\square$

**Operator Update 7** (Updates on D2). *Assume $\mathbf{D2}(x, y, C; E)$ is false and becomes true after an update involving $(a, b)$. Then,*

- *if the update is $U1 : (a \quad b) \rightsquigarrow (a - b)$ then $\begin{cases} b = y \\ a \in \mathrm{Ad}(x) \end{cases}$ or $\begin{cases} b = x \\ a \in \mathrm{Ne}(y) \end{cases}$ or $\begin{cases} a = y \\ b \in \mathrm{Ad}(x) \end{cases}$ or $\begin{cases} a = x \\ b \in \mathrm{Ne}(y) \end{cases}$,*

- *if the update is $U2 : (a \quad b) \rightsquigarrow (a \rightarrow b)$ then $\begin{cases} b = x \\ a \in \mathrm{Ne}(y) \end{cases}$ or $\begin{cases} a = x \\ b \in \mathrm{Ne}(y) \end{cases}$,*

- *the update cannot be $U3 : (a - b) \rightsquigarrow (a \quad b)$,*

- *the update cannot be $U4 : (a - b) \rightsquigarrow (a \rightarrow b)$,*

- *the update cannot be $U5 : (a \rightarrow b) \rightsquigarrow (a \quad b)$,*

- *if the update is $U6 : (a \rightarrow b) \rightsquigarrow (a - b)$ then $\begin{cases} a = y \\ b \in \mathrm{Ad}(x) \end{cases}$ or $\begin{cases} b = y \\ a \in \mathrm{Ad}(x) \end{cases}$,*

- *the update cannot be $U7 : (a \rightarrow b) \rightsquigarrow (a \leftarrow b)$.*

*Proof.* According to Lemma 5, either $\mathrm{Ne}^u(y)$ gained an element that was in $\mathrm{Ad}(x)$, or $\mathrm{Ad}^u(x)$ gained an element that was in $\mathrm{Ne}(y)$. We now study the necessary conditions for each update, if it was applied and made $\mathbf{D2}(x, y, C; E)$ become true.

- U3, U4, U5 and U7 do not add any element to $\mathrm{Ne}^u(y)$ or $\mathrm{Ad}^u(x)$. So they cannot make $\mathbf{D2}(x, y, C; E)$ become true.

- U1's only modifications to sets $\mathrm{Ne}(y')$ and $\mathrm{Ad}(x')$ are to add $a$ to $\mathrm{Ne}^u(b)$, add $a$ to $\mathrm{Ad}^u(b)$, and same exchanging $a$ and $b$. So by Lemma 5, $\begin{cases} b = y \\ a \in \mathrm{Ad}(x) \end{cases}$ or $\begin{cases} b = x \\ a \in \mathrm{Ne}(y) \end{cases}$ or $\begin{cases} a = y \\ b \in \mathrm{Ad}(x) \end{cases}$ or $\begin{cases} a = x \\ b \in \mathrm{Ne}(y) \end{cases}$.

- U2 's only modifications to sets $\mathrm{Ne}(y')$ and $\mathrm{Ad}(x')$ are to add $b$ to and $\mathrm{Ad}^u(a)$ and same exchanging $a$ and $b$. So by Lemma 5 $\begin{cases} a = x \\ b \in \mathrm{Ne}(y) \end{cases}$ or $\begin{cases} b = x \\ a \in \mathrm{Ne}(y) \end{cases}$.

- U6's only modifications to sets $\mathrm{Ne}(y')$ and $\mathrm{Ad}(x')$ are to add $a$ to $\mathrm{Ne}^u(b)$ and same exchanging $a$ and $b$. So by Lemma 5 $\begin{cases} b = y \\ a \in \mathrm{Ad}(x) \end{cases}$ or $\begin{cases} a = y \\ b \in \mathrm{Ad}(x) \end{cases}$.

$\square$

### B.5.8 D3

**Operator Update 8** (Updates on D3). *Assume $\mathbf{D3}(x, y, C; E)$ is false and becomes true after an edge update involving $(a, b)$. Also assume that $\mathbf{D2}(x, y, C; E)$ holds true after the edge update, then*

- *if the update is $U1 : (a \quad b) \rightsquigarrow (a - b)$ then $\{a, b\} \subset \mathrm{Ne}^u(y) \cap \mathrm{Ad}^u(x)$,*
- *if the update is $U2 : (a \quad b) \rightsquigarrow (a \rightarrow b)$ then $\{a, b\} \subset \mathrm{Ne}^u(y) \cap \mathrm{Ad}^u(x)$,*
- *the update cannot be $U3 : (a - b) \rightsquigarrow (a \quad b)$,*
- *the update cannot be $U4 : (a - b) \rightsquigarrow (a \rightarrow b)$,*
- *the update cannot be $U5 : (a \rightarrow b) \rightsquigarrow (a \quad b)$,*
- *the update cannot be $U6 : (a \rightarrow b) \rightsquigarrow (a - b)$,*
- *the update cannot be $U7 : (a \rightarrow b) \rightsquigarrow (a \leftarrow b)$.*

*Proof.* We recall that $\mathbf{D3}(x, y, C; E)$ is: $C$ is a clique. Assume $\mathbf{D3}(x, y, C; E)$ is false and becomes true after an update involving $(a, b)$. Similarly to Lemma 2, since $C$ is not changed by an edge update, the only way for $\mathbf{D3}(x, y, C; E)$ to become true is for the edge update to connect two nodes in $C$ that were not adjacent before. Only U1 and U2 render two nodes adjacent, namely $a$ and $b$, so they are the only updates that can make $\mathbf{D3}(x, y, C; E)$ become true.

For U1 and U2, we have: $\{a, b\} \subset C$. If we further assume that $\mathbf{D2}(x, y, C; E)$ holds true after the edge update, then $\{a, b\} \subset C \subset \mathrm{Ne}^u(y) \cap \mathrm{Ad}^u(x)$. $\qquad\square$

### B.5.9 D4

**Operator Update 9** (Updates on D4). *Assume $\mathbf{D4}(x, y, C; E)$ is false and becomes true after an edge update involving $(a, b)$. Then,*

- *the update cannot be $U1 : (a \quad b) \rightsquigarrow (a - b)$,*
- *if the update is $U2 : (a \quad b) \rightsquigarrow (a \rightarrow b)$ then $y = b$,*
- *the update cannot be $U3 : (a - b) \rightsquigarrow (a \quad b)$,*
- *if the update is $U4 : (a - b) \rightsquigarrow (a \rightarrow b)$, then $y = b$,*
- *if the update is $U5 : (a \rightarrow b) \rightsquigarrow (a \quad b)$, then $y = b$,*
- *if the update is $U6 : (a \rightarrow b) \rightsquigarrow (a - b)$ then $y = b$,*
- *if the update is $U7 : (a \rightarrow b) \rightsquigarrow (a \leftarrow b)$ then $y \in \{a, b\}$.*

*Proof.* Recall that $\mathbf{D4}(x, y, C; E)$ is: $E = C \cup \mathrm{Pa}(y)$. Since $C$ and $E$ do not change, the only way for $\mathbf{D4}(x, y, C; E)$ to become true is for the edge update to change $\mathrm{Pa}(y)$. The only updates that can change $\mathrm{Pa}(y)$ are U2, U4, U5, U6, and U7, when $y$ is $b$, or U7 when $y$ is $a$ or $b$. $\qquad\square$

### B.5.10 R1 to R6

The reverse operators are very similar to the insert operators, and the necessary conditions can be adapted. We add them to Table 4.

## B.6 ISSUE WITH FAST GES

We found an issue with the Fast GES algorithm that may explain its degraded performance compared to the GES algorithm.

During its efficient update of the operators, fGES computes the score of all possible operators Insert$(x, y, T)$ for a pair of nodes $x$ and $y$, but only saves the Insert with the highest score. However, this insert might not be a valid operator (for example, it might not satisfy the I3 constraint). Meanwhile, another Insert$(x, y, T')$ for the same pair of nodes $x$ and $y$ might be valid and have the highest score of all valid operators. Such an operator would be missed by fGES.