# OpenReview forum: "Extremely Greedy Equivalence Search"
_auai.org/UAI/2024/Conference — UAI 2024 spotlight_

### Official Review · Reviewer_JD2X · 2024-03-05

**Q2-1 Originality-Novelty:** 3
**Q2-2 Correctness-Technical Quality:** 4
**Q2-5 Clarity Of Writing:** 3

**Q10 Ethical Concerns:**

No.

**Q1 Summary And Contributions:**

The paper takes Greedy Equivalence Search (GES), one of the foundational causal discovery algorithms, and proposes an improved eXtremely Greedy Equivalence Search (XGES). The authors provide formal proof for the correctness of the algorithm and show that it dominates GES with a comprehensive simulation study. Furthermore, the authors also detail how to implement it efficiently by reworking its algorithmic formulation, showing in the process that it is faster and more accurate than GES.

**Q2-3 Extent To Which Claims Are Supported By Evidence:**

4: Excellent: all claims are supported by very convincing evidence (in the form of comprehensive experimental evaluation, rigorous mathematical proofs, detailed (pseudo-)code, precise references, well-motivated and realistic assumptions) and the authors deliver what they promise.

**Q2-4 Reproducibility:**

4: Excellent: key resources (e.g. proofs, code, data) are available and key details (e.g. proof sketches, experimental setup) are comprehensively described for competent researchers to confidently and easily reproduce the main results.

**Q3 Main Strengths:**

The paper thoroughly proves that XGES is theoretically sound (with formal proofs), that it performs better in practice (with a thorough simulation study), and that it can be implemented efficiently (working with computational complexity as well as empirical timings). The topic's treatment is complete and thorough, and the writing is quite clear. The topic itself is highly relevant.

**Q4 Main Weakness:**

While the paper is clear in its exposition, it reads like two papers that were written independently and then collated together without adaptation. Starting from page 6, many quantities defined earlier in the paper are defined again but in somewhat different ways, confusing readers who are not entirely familiar with the topic and who are reading the paper for the first time.

**Q5 Detailed Comments To The Authors:**

Here are more detailed comments with page references.

* abstract: "It has strong theoretical guarantees on infinite data but can fail in practice on finite data." This statement sounds like it is specific to GES, but it is true for all causal discovery algorithms.
* page 1: "In particular, it is not practical for graphs with a non-trivial number of edges, e.g., more edges than nodes." Taken literally, this would mean we can only learn trees (not even polytrees), which is obviously not the case in the literature.
* page 1: "And so we cannot apply GES to the kinds of large-scale problems that we regularly encounter in machine learning." The authors should qualify this statement better; GES is reasonably efficient and still usable with a few hundred nodes. Causal DAGs larger than that quickly become impossible to inspect visually and reason about, limiting their usefulness, in my opinion.
* page 2: "These methods can model causal relations using neural networks", but neither of the references does.
* page 2: "The goal of causal discovery is to recover the graph G from the joint distribution p or from samples drawn from p". All causal discovery algorithms I am familiar with do the latter. Could the author provide an example of the former?
* page 3: "The score used by GES is the Bayesian Information Criterion (BIC)." This is usually the case, but I do not believe GES requires it. Could the authors confirm?
* page 3: "The original BIC has alpha = 1." This is not correct as written. Different values of alpha give different penalised likelihood scores (AIC is another example), which have vastly different interpretations and properties, as opposed to slight variations of BIC.
* page 3: "GES chooses the model class of Gaussian linear models." I do not believe that is a requirement of GES, which is also often applied to categorical data.
* page 4: "Example of failure." This example is somewhat problematic because it compares models with very different numbers of parameters but learned from the same sample size and, therefore, with very different accuracies. For instance, d = 25 and pd = 2d implies 10000 observations/100 parameters = 100, while d = 50 and pd = 4d implies 10000 observations/350 parameters = 30. I think this can introduce quite some confounding in the interpretation of the example.
* page 4: "A negative DeltaS indicates that GES failed..." The authors seem to imply that the model with the highest possible score for a data set must be the generating model, which I do not think is the case for finite samples. I believe this point should be clarified. See the paper "On the pitfalls of Gaussian likelihood scoring for causal discovery" on this topic.
* page 5: t is an undefined quantity in Algorithm 2.
* page 5: the descriptions of XGES-0 (Section 3.2) and XGES (Section 3.3) look identical to those of greedy search without and with a tabu list. I would like the authors to differentiate those descriptions from those of hill climbing and tabu search to help the reader appreciate how they differ.
* page 5: "MECs are sets of DAGs whose size can grow exponentially with the number of nodes." S. Gillispie and M. Perlman (2002) have shown that while the number of equivalence classes is of the same order of magnitude as the space of the DAGs, most contain few DAGs, and as many as 27.4% contain just a single DAG. Could the authors clarify how these two statements should be reconciled?
* page 7: "It is more efficient to leave them and discard them when choosing the best operator" Could the authors clarify why in the text?
* page 8: "We report the F1 score," but I believe reporting SID in a causal discovery setting would have been more informative.
* page 8: "Robustness to the Sample Size." I might be missing something here. Isn't it by construction that the penalty in BIC loses power as the sample size increases? And BIC is consistent on complete data, so it is supposed to return the true model as asymptotic sample sizes. A similar point could be made about the following paragraph.
* page 8: the Conclusions are too short to be informative.

**Q9 Complying With Reviewing Instructions:**

Yes

---

> ### Author Rebuttal · Authors · 2024-04-03
>
> Thank you very much for your thorough review and comments, they improved our paper. We answer your questions below.
>
> > While the paper is clear in its exposition, it reads like two papers that were written independently and then collated together without adaptation. [...]
>
> We understand your point but we believe this structure also has its advantages. The original GES paper is 30 pages long and explaining it at the same time as presenting XGES was indeed a challenge. We found that dividing our paper with more high level concepts first, and more technical implementation details later was easier to follow. To help ease the transition between the two parts, we have now added this: *“In section 2.3, we reviewed how GES theoretically navigates the space of MECs by applying operators to MECs and by computing the score of MECs. We now review how GES implements each of these MECs' manipulations in practice, and then show how to make them more efficient.”*
>
>
> > page 1:[...] Taken literally, this would mean we can only learn trees [...]
>
> We agree with you. We now say: *"In particular, its performance decreases as graphs get more complex , e.g. more than two parents per node.”* (Indeed, experiments showed that increasing the density of edges of the ground truth graph hurts GES (more than it hurts XGES).)
>
> > page 1: [...] GES is reasonably efficient and still usable with a few hundred nodes.
>
> We agree that GES works reasonably well with a few hundred nodes, but mostly when the target graph is relatively sparse. The experiments show that once the density increases, then GES is hurt and XGES has an advantage.
>
> > page 2: [...], but neither of the references does.
>
> The second reference does (Brouillard et al., 2020, abstract "method based on neural networks"). The first one has been adapted to do so in follow-up works that we now cite.
>
> > page 2: [...] Could the author provide an example of the former?
>
> Thank you for the question. Indeed, the latter makes more sense, we have now removed the former.
>
> > page 3: [...] This is usually the case, but I do not believe GES requires it. [...]
>
> You are correct, it is not a requirement. Thank you for ensuring the text is precise. We updated the text to: *"A score commonly used by GES is the Bayesian Information Criterion (BIC)."*
>
> > page 3: "The original BIC has alpha = 1." This is not correct as written.
>
> We believe it is correct that the BIC corresponds to our formula with  $\alpha=1$ ($\alpha$ is multiplied by $\log(n)/2$ in the equation).
>
> > page 3: [...] I do not believe that is a requirement of GES, which is also often applied to categorical data.
>
> We agree with you. We now write *“A common model class used for the BIC with continuous data is the class of Gaussian linear models, [...]”* and later on *“More generally, the BIC is locally consistent for most model classes of the exponential family (e.g. the multinomial distribution for discrete data) [Chickering 2002b]."*
>
> > page 4: [...] I think this can introduce quite some confounding in the interpretation of the example.
>
> We understand your concern but the point of the figure is to compare models against each other, not necessarily across regimes. In fact, this shows the robustness of the results for heterogeneous sample/parameter ratios.
>
> > page 4: [...] The authors seem to imply that the model with the highest possible score for a data set must be the generating model, [...]
>
> We did not imply such a statement (at least we did not mean to). We believe our sentence is correct. For e.g., we did not write “higher than any $M$”. We clarified the sentence by adding: “(note that it does not imply $G^*$ is the global maximizer).” Is it clearer?
> We also include your reference in the related work since its conclusions about Gaussian scoring and GES are directly relevant.
>
> > page 5: t is an undefined quantity
>
> Thank you, we removed it.
>
> > page 5: the descriptions of XGES-0 [...]
>
> Thank you for the suggestions. We updated the text to discuss tabu search, it has some connections but it is different. For e.g., XGES does not maintain any memory structure as tabu search does (the tabu list).
>
> > page 5: "[...] Could the authors clarify how these two statements should be reconciled?
>
> We believe our statement is correct (even though not common). E.g., consider the DAGs with complete skeleton: there are $d!$ of them (one per topological sort of the nodes) but they all belong to the same MEC. $d!$ grows super exponentially in $d$. So it seems correct to say the size **can** grow exponentially with $d$. Do you agree?
>
> > page 7: [...] Could the authors clarify why in the text?
>
> We now say *“In practice, it is more efficient to leave the invalid operators in the list and only check the validity of an operator before applying it (and discarding it if invalid). Indeed, if we recheck the validity of all operators after each operation, a single operator will be rechecked at each step until it is applied instead of being checked only once.”*

---

### Official Review · Reviewer_rwr4 · 2024-03-18

**Q2-1 Originality-Novelty:** 2
**Q2-2 Correctness-Technical Quality:** 3
**Q2-5 Clarity Of Writing:** 3

**Q1 Summary And Contributions:**

This paper considers score-based structure learning in Bayesian networks. It presents Extremely Greedy Equivalence Search (XGES), a variant of the well-known Greedy Equivalence Search (GES) algorithm.

**Q2-3 Extent To Which Claims Are Supported By Evidence:**

2: Fair: the main claims are somewhat supported by evidence (but the experimental evaluation may be weak, or does not match entirely with the claims, important baselines may be missing, proofs contain important ideas but lack rigor, algorithmic details are only discussed superficially, references are imprecise, assumptions are not sufficiently motivated or explicated, etc.).

**Q2-4 Reproducibility:**

3: Good: key resources (e.g. proofs, code, data) are available and key details (e.g. proofs, experimental setup) are sufficiently well-described for competent researchers to confidently reproduce the main results.

**Q3 Main Strengths:**

1) The paper present a reasonable approach to solve an important problem.

**Q4 Main Weakness:**

1) The paper has limited novelty. The idea of XGES is not new. Already Chickering [2002b] tested a similar approach under the name OPS (see page 534 of his paper). OPS did not have reversals as XGES but Chickering proposed also that idea in the discussion of his paper.

Chapter 4 has novelty. However, it is not clear what is the actual impact of the proposed improvements.


2) The claims of that XGES is better than GES are not supported by the experiments. Specifically, the chosen performance measures are not the best ones for the task.

First of all, score-based structure learning tries to find a DAG that maximises a score that how well a DAG fits to the data. With specific scores and under specific assumptions (including infinite data), that DAG happens to coincide with the true causal graph but there is nothing inherently causal in score-based structure learning.

As the objective of a score-based method is to find a high-scoring DAG, a natural way to compare score-based methods is to compare the scores of the found DAGs.

Comparing the found DAG to the data-generating DAG muddies the waters. Now the method that finds the highest scoring structure might perform worse because the data-generating DAG is not necessarily the one that maximises the score.

Thus, currently we do not know how much of the performance differences are due to differences between GES and XGES and much due to "deficiencies" of the scoring criterion.

Second, the paper claims that XGES is 10 times faster than GES. However, the GES implementations used R and Java while XGES was implemented using C++. So the observed difference might be simply because C++ programs tend to be faster than R/Java programs.

To give a reliably estimate on running times one should either make comparable implementations of all methods or consider a proxy that does not depend on implementation details. For example, one could count how many times each algorithm calls the scoring function S.

**Q5 Detailed Comments To The Authors:**

Algorithm 2 has a variable t that is set to 0 and never incremented. It only appears in a comparison with >= -t. Why not use >= 0?

Theorem 1: For any alpha > 0, ...

**Q9 Complying With Reviewing Instructions:**

Yes

---

> ### Author Rebuttal · Authors · 2024-04-03
>
> Thank you for your review and the comments, they have improved the paper. We discuss your concerns below. We believe we addressed them.
>
> > The paper has limited novelty. The idea of XGES is not new. Already Chickering [2002b] tested a similar approach under the name OPS [...]
>
> Thank you very much for bringing this approach to our attention. It nicely introduces our work. We now include it in our related work and discussion. However, XGES is different from OPS, we disagree it has limited novelty:
> - **XGES-0 is different from OPS.** Both XGES-0 and OPS consider all operators simultaneously, but only XGES-0 has reversals, and only XGES-0 favors deletions (and reversal) over insertions.
> - **XGES is not XGES-0.** The review seems to ignore section 3.3 about the main algorithm. We believe that XGES is another contribution.
> - **OPS was not studied and its advantages not discovered.** In Chickering [2002b], OPS seems to make no improvement compared to GES, and was not studied further. In our paper, we show (among other results) that variants of OPS actually have value. We believe that these findings are important for the field.
>
> Overall, Chickering proposed excellent future work directions, and we totally agree that XGES is building on them. We clarified how we build on them in the related work section and we maintain that XGES is a valuable and novel contribution.
>
> For completeness, we added OPS in the experiments (see Figure 4 at https://anonymous.4open.science/r/uai-rebuttal-C5A3/xges-rebuttal-figures.pdf). We see that **without XGES-0 improvements, OPS does not improve on GES**, as found by Chickering [2002b].
> This is expected because Delete operations can only increase the score by no more than $\alpha\log(n)/2$ (the BIC penalty) as it will always hurt the likelihood to delete edge and it will only improve the regularization by $\alpha\log(n)/2$. So Deletions can have by design only a very small positive score. So even with OPS design considering all operators, Deletions will likely be applied after most inserts, not changing the search of GES. We added this discussion in the main text.
>
> > Chapter 4 has novelty. However, [...]  the chosen performance measures are not the best ones for the task. [...] a natural way to compare score-based methods is to compare the scores of the found DAGs. Comparing the found DAG to the data-generating DAG muddies the waters. [...] Thus, currently we do not know how much of the performance differences are due to differences between GES and XGES and much due to "deficiencies" of the scoring criterion.
>
> Thank you for your detailed comment. We understand your concerns but strongly disagree with your conclusions.
>
> - We agree that maximizing the score is not always equivalent to minimizing SHD. This was the focus of our section 3.1, and it served as the motivation for XGES-0. We did not claim otherwise.
> - To clear any doubt about our metrics **we now report the BIC score achieved by all methods alongside SHD.** (See Figure 2 at https://anonymous.4open.science/r/uai-rebuttal-C5A3/xges-rebuttal-figures.pdf). **The conclusions are identical.** XGES reaches scores better than other methods. The performance ordering of the method is the same: XGES > XGES-0 > GES-r > GES > fGES.
>
> To build on your astute comment, we add a figure showing correlation between score and SHD (see Figure 1 (left) at https://anonymous.4open.science/r/uai-rebuttal-C5A3/xges-rebuttal-figures.pdf). This new plot reveals that score and SHD are strongly correlated. Hence, the score is a good proxy for SHD, and vice versa: it is not deficient.
>
> In summary, we disagree that our performance measures were inappropriate, and we showed that we reach identical conclusions with your suggested metrics. In addition, we provide experiments showing that score and SHD are strongly correlated. We conclude that the performance differences **are** due to differences between GES and XGES and **not** due to "deficiencies" of the scoring criterion.
>
> > Second, the paper claims that XGES is 10 times faster than GES. [...]
>
> Thank you for the legitimate question and suggestions. We emphasize that **GES is implemented as a C++ library** with R bindings (the sources of the R `pcalg` package with the `gies.cpp` file at https://github.com/cran/pcalg/tree/master/src.)
> So we believe the comparison to GES is fair. We agree however that the comparison to fGES in Java, might be biased due to the programming language. We added this discussion to the paragraph about running time.
>
> > [...] For example, one could count how many times each algorithm calls the scoring function S.
>
> Great suggestion, thank you. We generated a figure equivalent to Figure 4 (right) for the best performing methods (GES, GES-r and XGES-0, XGES), this time with the number of computations of the BIC score instead of the total time (Figure 3 at https://anonymous.4open.science/r/uai-rebuttal-C5A3/xges-rebuttal-figures.pdf). We observe similar conclusions.

---

### Official Review · Reviewer_DvPg · 2024-03-22

**Q2-1 Originality-Novelty:** 3
**Q2-2 Correctness-Technical Quality:** 3
**Q2-5 Clarity Of Writing:** 4

**Q1 Summary And Contributions:**

GES is a popular causal discovery algorithm because it can be executed in a greedy manner while retaining its theoretical guarantee. This work identifies some of GES's failure on finite samples and and shows that it can be blocked in local minima. The authors then propose a new heuristic to improve the search procedure of GES and show that it retains the theoretical guarantee. The heuristic involves favoring deleting edges early in the search over inserting edges. An efficient implementation is provided, and experiment results demonstrate that the method is fast and performs well.

**Q2-3 Extent To Which Claims Are Supported By Evidence:**

3: Good: the main claims are supported by convincing evidence (in the form of adequate experimental evaluation, proofs, (pseudo-)code, references, assumptions).

**Q2-4 Reproducibility:**

3: Good: key resources (e.g. proofs, code, data) are available and key details (e.g. proofs, experimental setup) are sufficiently well-described for competent researchers to confidently reproduce the main results.

**Q3 Main Strengths:**

GES is a popular causal discovery algorithm, and finding ways to improve its search procedure is highly relevant and important in the causal discovery community.

The heuristic leads to both improved reliability and efficiency over GES, while retaining the theoretical guarantee. The theoretical results and resulting algorithm appear to be sound, although I did not check the proofs carefully.

**Q4 Main Weakness:**

>GES considers insertions, deletions, and optionally reversals in three separate phases. Rather, we consider all operations simultaneously, where deletions, insertions, and reversals can interleave in any order

Although the original GES indeed consists of a forward and a backward phase, there are other greedy DAG search methods that consider all three operations simultaneously, such as the method in "Identifiability of Gaussian structural equation models with equal error variances" (although the greedy method is for equal variance setting and without theoretical result). Therefore, the XGES-0 method may not be completely novel.

Another weakness is that the implementation of the proposed XGES is in C++, while the original GES is implemented in Java. Therefore, one cannot be sure that the running time is improved because of the heuristic, or because of a different (perhaps more efficient) programming language. In this case, apart from running time, would it help to also report how many times the BIC score is computed?

**Q5 Detailed Comments To The Authors:**

- Is there a better to visualize the left panel in Figure 4? The scatter plot makes it difficult to compare across methods.

**Q9 Complying With Reviewing Instructions:**

Yes

---

> ### Author Rebuttal · Authors · 2024-04-03
>
> Thank you very much for your review and comments, they improved our paper. We answer your questions and concerns below.
>
>
> &nbsp;
>
> ---
>
> > Although the original GES indeed consists of a forward and a backward phase, there are other greedy DAG search methods that consider all three operations simultaneously, such as the method in "Identifiability of Gaussian structural equation models with equal error variances" (although the greedy method is for equal variance setting and without theoretical result). Therefore, the XGES-0 method may not be completely novel.
>
> Thank you for the great reference, we added it to our discussion of related work. We agree other methods revolve around similar ideas. We believe that studying and highlighting the superiority of such approaches for GES, as we do with XGES-0, is a contribution.
> We have clarified the related work with your reference.
>
> We hope that you will also believe that XGES-0 and our analysis of it are a valuable contribution to the community. In addition, XGES-0 is complemented and improved with the XGES method.
>
> &nbsp;
>
> ---
>
> > Another weakness is that the implementation of the proposed XGES is in C++, while the original GES is implemented in Java. Therefore, one cannot be sure that the running time is improved because of the heuristic, or because of a different (perhaps more efficient) programming language.
>
> Thank you for the legitimate question and the great suggestion. We emphasize that **GES baselines are not implemented in Java. GES is actually implemented as a C++ library** with R bindings. (Sources of the R `pcalg` package with the `gies.cpp` file are at https://github.com/cran/pcalg/tree/master/src.)
> So we believe the comparison to GES is fair.
>
> We agree however that the comparison to fGES, which uses Java, might be biased due to the programming language. We added this discussion to the paragraph about running time.
>
> &nbsp;
>
> ---
>
> > In this case, apart from running time, would it help to also report how many times the BIC score is computed?
>
> As suggested, we investigated how many times the BIC score is computed for the best performing methods (GES and GES-r). We generated a figure equivalent to Figure 4 (right), but this time with the number of computations of the BIC score instead of the total time (see Figure 3 in https://anonymous.4open.science/r/uai-rebuttal-C5A3/xges-rebuttal-figures.pdf). We observe similar conclusions.
>
>
> &nbsp;
>
> ---
>
> > Is there a better to visualize the left panel in Figure 4? The scatter plot makes it difficult to compare across methods.
>
> That is a good point, thank you. We agree that the scatter plot was not the best visualization. Instead, we now generate box-plots similar to Figure 2 and Figure 7, using the recall and precision metrics. See link Figure 5 at https://anonymous.4open.science/r/uai-rebuttal-C5A3/xges-rebuttal-figures.pdf.
>
> —
>
> Thank you again for the review. The updated figures requested by you or the other reviewers are available at https://anonymous.4open.science/r/uai-rebuttal-C5A3/xges-rebuttal-figures.pdf.

---

### Official Review · Reviewer_Mbyh · 2024-03-23

**Q2-1 Originality-Novelty:** 3
**Q2-2 Correctness-Technical Quality:** 3
**Q2-5 Clarity Of Writing:** 3

**Q1 Summary And Contributions:**

This paper proposed an altered version of GES that chooses between more operations when considering what CPDAG to move to next. The proposed algorithm is no longer limited to a forward phase and a backward phase, but is now allowed to add, delete, and reverse (legal) edges at every step. The modification retains GES's asymptotic guarantees. The authors justify the need for this alteration by showing that GES fails on larger denser models. The method is implemented in C++ and a python wrapper is provided. The authors compare their algorithm against other versions of GES and show superior performance in terms of correctness and computation time.

**Q2-3 Extent To Which Claims Are Supported By Evidence:**

3: Good: the main claims are supported by convincing evidence (in the form of adequate experimental evaluation, proofs, (pseudo-)code, references, assumptions).

**Q2-4 Reproducibility:**

3: Good: key resources (e.g. proofs, code, data) are available and key details (e.g. proofs, experimental setup) are sufficiently well-described for competent researchers to confidently reproduce the main results.

**Q3 Main Strengths:**

The algorithm has stellar performance and scales incredibly well. The algorithm has a Python wrapper so it should be easy for practitioners to use for their analyses.

**Q4 Main Weakness:**

I would like to see how XGES compares to other state of the art causal discovery algorithms. That being said, I do understand the logic behind only comparing against other versions of GES since this is a modification of GES.

10,000 samples is a lot---many real world data have less than 1,000 samples. There is a set of simulations that explore smaller sample sizes, but the simulations that explore other aspects off the algorithmic performance all use 10,000 samples.

**Q5 Detailed Comments To The Authors:**

I really enjoyed this paper and think it is really solid work. Accordingly, I only have a few minor comments/questions:

You say: "We investigate GES failure. Is the issue that GES cannot reach the global maximizer with greedy search? If so, changing the search heuristic could help bypass local maxima. Or is it that GES effectively finds the global maximizer, but this maximizer is not the true graph?" and "we hypothesize that over inserting may lead GES into local maxima before the second phase can correct it."

Does this mean that the local maxima reached in the first phase fails to satisfy the causal Markov condition? That is, insufficient edges have been added to represent the distribution?

In your simulations, what was the reason for only sampling positive betas (for the majority of your experiments?)

In Figure 4, what is meant by precision and recall? For example, are these adjacency statistics, or are you taking edge direction into account as well?

Also in Figure 4, it is interesting to see that XGES has a better running time than existing GES variants. How much of this do you think is due to implementing the algorithm in C++ and how much of this do you think is due to the other variants wasting a lot of time adding all those extra edges?

It is interesting that XGES is not much slower than XGES-0 even though XGES is effectively running XGES-0 over and over again. Can you comment on this? Anecdotally, do you know about how many times XGES-0 is rerun by XGES before converging?

**Q9 Complying With Reviewing Instructions:**

Yes

---

> ### Author Rebuttal · Authors · 2024-04-03
>
> Thank you for the precise review and encouraging comments, they definitely helped improve our paper. We respond to your comments below.
>
> &nbsp;
>
> ---
>
> > 10,000 samples is a lot---many real world data have less than 1,000 samples. There is a set of simulations that explore smaller sample sizes, but the simulations that explore other aspects off the algorithmic performance all use 10,000 samples.
>
> Thank you for the comment, it is a good point. We observe similar behavior for different sample sizes. As you pointed out, we studied the effect of sample size in Figure 5, and the conclusion were that sample size can shift the performance of all methods up or down, but did not change the ordering of method performances.
>
>
> &nbsp;
>
> ---
>
> > Does this mean that the local maxima reached in the first phase fails to satisfy the causal Markov condition? That is, insufficient edges have been added to represent the distribution?
>
> Thank you for your question. Our understanding is that indeed, the local maxima fails to satisfy the causal Markov conditions. But mostly because incorrect edges were added during the forward search, inducing incorrect conditional dependencies.
>
>
> &nbsp;
>
> ---
>
> > In your simulations, what was the reason for only sampling positive betas (for the majority of your experiments?)
>
> Thank you for your question. We originally sampled betas away from zero to ensure faithfulness, and when doing so we sampled positive betas, there was no other reason. But since it was also possible to obtain faithful data with negative beta, we included this experiment in Figure 10 to study the robustness to another data generative process.
>
>
>
> &nbsp;
>
> ---
>
> > In Figure 4, what is meant by precision and recall? For example, are these adjacency statistics, or are you taking edge direction into account as well?
>
> Thank you for this important question. Yes, we take edge direction into account: an incorrect direction will hurt precision and recall. We now detail the metric in Appendix, especially how to handle Markov Equivalence Classes, which can contain undirected edges.
>
>
> &nbsp;
>
> ---
>
> > Also in Figure 4, it is interesting to see that XGES has a better running time than existing GES variants. How much of this do you think is due to implementing the algorithm in C++ and how much of this do you think is due to the other variants wasting a lot of time adding all those extra edges?
>
> Thank you for the great question. We would like to highlight that even though GES is called from R code, the core of the GIES method in R is implemented in C++. (See the sources of the R `pcalg` package with the `gies.cpp` file at https://github.com/cran/pcalg/tree/master/src.) Hence, we believe the programming language did not have an influence for comparing XGES against GES and GES-r. Regarding fGES, we agree that Java likely has some overhead. We now include this discussion in the paper.
>
> &nbsp;
>
> ---
>
> > It is interesting that XGES is not much slower than XGES-0 even though XGES is effectively running XGES-0 over and over again. Can you comment on this? Anecdotally, do you know about how many times XGES-0 is rerun by XGES before converging?
>
> Thank you for the astute question. After XGES deletes an edge, it does not run XGES-0 from scratch, instead, it resumes XGES-0 on the resulting graph. In most cases, XGES-0 terminates only after a few steps. We included this discussion in the paper.
> In a new figure (Figure 3 at https://anonymous.4open.science/r/uai-rebuttal-C5A3/xges-rebuttal-figures.pdf), we show the number of calls to the scoring function made by each algorithm. We see that XGES makes more BIC score evaluation than XGES-0 but still less than GES.
>
> —
>
> Thank you again for the review. The updated figures requested by you or the other reviewers are available at https://anonymous.4open.science/r/uai-rebuttal-C5A3/xges-rebuttal-figures.pdf.

---

### Official Review · Reviewer_tapy · 2024-03-23

**Q2-1 Originality-Novelty:** 4
**Q2-2 Correctness-Technical Quality:** 3
**Q2-5 Clarity Of Writing:** 4

**Q10 Ethical Concerns:**

No.

**Q1 Summary And Contributions:**

This work provides a heuristic modification of the well-known GES algorithm in causal discovery. While preserving the asymptotic property of GES, the proposed algorithm greatly improves GES in terms of time and performance.

**Q2-3 Extent To Which Claims Are Supported By Evidence:**

4: Excellent: all claims are supported by very convincing evidence (in the form of comprehensive experimental evaluation, rigorous mathematical proofs, detailed (pseudo-)code, precise references, well-motivated and realistic assumptions) and the authors deliver what they promise.

**Q2-4 Reproducibility:**

4: Excellent: key resources (e.g. proofs, code, data) are available and key details (e.g. proof sketches, experimental setup) are comprehensively described for competent researchers to confidently and easily reproduce the main results.

**Q3 Main Strengths:**

With all the nicely illustrated comparisons, the simulation studies provided strong and robust evidence that XGES (and XGES-0) outperforms other popular variants of GES, particularly fGES. While GES is known to be sensitive to almost violations of faithfulness in finite samples (i.e., easily trapped by local minima, especially in the forward phase), XGES provides a good solution to this practical problem while preserving all the asymptotic properties of GES. Pseudocodes and implementation details are clear so readers can reliably reproduce their results.

**Q4 Main Weakness:**

I don't see any obvious weakness in this work. But if I had to mention one, the authors may want to consider other score-based algorithms (e.g., the Sparsest Permutation algorithm and its greedy variants) to have an even fuller comparison. By adopting the standard argument that unfaithfulness has a Lebesgue measure of zero, the main worry of causal discovery with finite samples is almost unfaithfulness. Different algorithms tried to solve this problem by proposing consistency assumptions weaker than faithfulness. So, it will be beneficial to see if XGES, similar to those algorithms, assumed a weaker-than-faithfulness assumption.

**Q5 Detailed Comments To The Authors:**

P.3 Even though your work assumes faithfulness, it would be clearer to state Chickering's results in Section 2.3 under faithfulness.

P.3 In the definition of BIC, I don't think you should use |G| to substitute the number of parameters of a given DAG G. It is correct for linear Gaussian models, but not for multinomial distributions (where BIC is still consistent).

P.3 It would be helpful to remind the readers about the sign of the BIC score when you define it. Also, it seems that you have used local minima (abstract) and local maxima (Section 3.1) at the same time to mean local optima.

P.5 What is t in Algorithm 2? I cannot see its function in the pseudocode.

P.8 It surprised me that fGES did not run for the denser simulation in Figure 4. Would you mind rerunning that and check if it is not a technical problem?

P.8 For the implementation speed comparison, it would be helpful to remind readers that the multiple algorithms are not compared with the same programming language (e.g., fGES in Java, GIES in R, and XGES in C++). This may explain the differences in time performance in Figure 4.

**Q9 Complying With Reviewing Instructions:**

Yes

---

> ### Author Rebuttal · Authors · 2024-04-03
>
> Thank you for the very thorough review, it definitely helped improve our paper. We respond to your comments below.
>
> &nbsp;
>
> ---
>
> > P.3 In the definition of BIC, I don't think you should use |G| to substitute the number of parameters of a given DAG G. It is correct for linear Gaussian models, but not for multinomial distributions (where BIC is still consistent).
>
> Thank you for the precision. We replaced it with $|\theta|$ and the sentence: *“[...] where $|\theta|$ is the number of model parameters (usually about the number of edges in $G$).”*
>
> &nbsp;
>
> ---
>
> > P.3 It would be helpful to remind the readers about the sign of the BIC score when you define it. Also, it seems that you have used local minima (abstract) and local maxima (Section 3.1) at the same time to mean local optima.
>
> Thank you very much for the suggestions. It should indeed be “local maxima”, we edited the paper accordingly.
>
> Regarding the sign of the BIC, we assume you are suggesting to remind the reader whether the BIC should ideally be maximized or minimized? Indeed, we don't think the sign of the BIC score is informative for continuous densities, it can be positive or negative, usually depending on how “spiked” is the density (equivalently p(x) can be > 1 for continuous densities). We adapted the text as follow: “The BIC is a model selection criterion trading off log-likelihood and model complexity. Models with higher BIC are preferred.”
>
> &nbsp;
>
> ---
>
> > P.5 What is t in Algorithm 2? I cannot see its function in the pseudocode.
>
> Thank you very much for pointing this out. The $t$ has indeed no purpose and we have now removed it.
>
> &nbsp;
>
> ---
>
> > P.8 It surprised me that fGES did not run for the denser simulation in Figure 4. Would you mind rerunning that and check if it is not a technical problem?
>
> Thank you for pointing this out. We also noticed it after submission and we have re-ran the experiment since. fGES runs on denser graphs, but takes more time than XGES. We updated the plots. (See Figure 1 (right) at https://anonymous.4open.science/r/uai-rebuttal-C5A3/xges-rebuttal-figures.pdf)
>
> &nbsp;
>
> ---
>
> > P.8 For the implementation speed comparison, it would be helpful to remind readers that the multiple algorithms are not compared with the same programming language (e.g., fGES in Java, GIES in R, and XGES in C++). This may explain the differences in time performance in Figure 4.
>
> This is a great point, thank you. We totally agree, we now remind readers about the different programming languages in the paragraph about speed comparison and in the Figure caption.
>
> Notably, we would like to highlight that even though GIES is called from R code, **the core of the GIES method in R is implemented in C++**. (See the sources of the R `pcalg` package with the `gies.cpp` file at https://github.com/cran/pcalg/tree/master/src.)
>
> Hence, we believe that the comparison between XGES and the GES variants were actually quite fair. Regarding fGES however, we agree it is likely that the Java implementation incurs some overhead. We added this discussion.
>
> &nbsp;
>
> —
>
> Thank you again for the review. The updated figures requested by you or the other reviewers are available at https://anonymous.4open.science/r/uai-rebuttal-C5A3/xges-rebuttal-figures.pdf.

---

### Meta-Review · Area_Chair_Ji1q · 2024-04-15

This paper proposes an algorithmic improvement on the GES algorithm for structure learning of Bayesian networks from data. The paper is well written and the authors have made an effort to make the proofs and code accessible to the reader. Most reviewers are positive and appreciate the authors' efforts to clear up their criticisms.